# A Variational Information Theoretic Approach to Out-of-Distribution Detection

**Sudeepta Mondal** [1]   **Zhuolin Jiang** [1]   **Ganesh Sundaramoorthi** [1]

## Abstract

We present a theory for the construction of out-of-distribution (OOD) detection features for neural networks. We introduce random features for OOD through a novel information-theoretic loss functional consisting of two terms, the first based on the KL divergence separates resulting in-distribution (ID) and OOD feature distributions and the second term is the Information Bottleneck, which favors compressed features that retain the OOD information. We formulate a variational procedure to optimize the loss and obtain OOD features. Based on assumptions on OOD distributions, one can recover properties of existing OOD features, i.e., shaping functions. Furthermore, we show that our theory can predict a new shaping function that out-performs existing ones on OOD benchmarks. Our theory provides a general framework for constructing a variety of new features with clear explainability.

## 1. Introduction

Machine learning (ML) systems are typically designed under the assumption that the training and test sets are sampled from the same statistical distribution. However, this often does not hold in practice. For example, during deployment, test data may include previously unseen classes. In such cases, the ML system may produce incorrect results with high confidence (DeVries & Taylor, 2018). Therefore, it is crucial to develop methods that enable ML systems to *detect out-of-distribution* (OOD) data. Detecting OOD data allows users to be alerted of potentially unreliable predictions and enables the system to adapt accordingly. OOD detection has gained considerable attention recently (Yang et al., 2022).

Recent state-of-the-art (SoA) (Sun et al., 2021; Djurisic et al., 2022; Ahn et al., 2023; Sun & Li, 2022; Zhao et al.,

2024; Xu et al., 2023; Zhang et al., 2024) has focused on identifying descriptors of the data that can distinguish between OOD and ID data. In particular, feature-shaping functions have been shown to be promising. In feature shaping, features, usually from the penultimate layer of a pre-trained network, are input to a *shaping* function and then used to score incoming data. Examples of these approaches include ReAct (Sun et al., 2021), FS-OPT (Zhao et al., 2024), VRA (Xu et al., 2023), ASH (Djurisic et al., 2022). While these approaches have provided advancements to SoA on several benchmarks, most of these are empirically driven rule-based methods, and may not generalize well over new unseen datasets (Zhao et al., 2024). It is thus beneficial to understand under what conditions that these methods will work so that we may understand when to employ one method over another. One way to do this is to develop a theory where one can derive features for OOD detection (henceforth called OOD features for brevity) as a function of underlying assumptions (e.g., statistical distributions). This could potentially offer a way to critically examine existing methods. It could also potentially enable the systematic development of new features that may generalize better.

Towards this goal, we develop a theory to formulate OOD features based on underlying statistical distributions of ID and OOD distributions. We develop a novel loss functional, based on information theory, defined on the set of OOD features whose optimization yields OOD features as a function of the underlying statistical distributions. Unlike current approaches, our OOD features are random and thus follow a statistical distribution. The mean value models the deterministic shaping features in the literature.

Our loss aims to determine the OOD feature that maximally separates resulting ID and OOD feature distributions through the Kullback-Leiber (KL) divergence. As separating distributions by itself is ill-posed, we propose a novel use of the Information Bottleneck (IB) (Tishby et al., 2000) as regularization. In our use, IB seeks compressed features that preserve the information the data has about OOD, aiming for a feature representation that contains only the information necessary for OOD detection. As this loss functional is defined on probability measures (representing the distribution of the OOD feature), it is an infinite dimensional optimization problem, and thus we use the calculus of variations (Troutman, 2012) to derive the optimization procedure. Our

[1]RTX Technology Research Center (RTRC), East Hartford, CT 06118. Correspondence to: Ganesh Sundaramoorthi <ganesh.sundaramoorthi@rtx.com>.

*Proceedings of the 42nd International Conference on Machine Learning*, Vancouver, Canada. PMLR 267, 2025. Copyright 2025 by the author(s).

theory offers an explanation of several techniques employed in SoA rule- based approaches, and suggests a new shaping function that out-performs other shaping functions in SoA.

There have been recent theories for OOD detection (Zhao et al., 2024; Xu et al., 2023). These works have introduced the novel idea of formulating OOD features through a loss function rather than empirically driven rule-based approaches of the past, and motivates our work. In contrast to the aforementioned works, our theory employs a novel information-theoretic loss function, which offers several advantages. Our theory shows how different assumptions on the OOD distribution lead to different OOD feature shaping approaches. Our theory is able to more accurately offer an explanation for properties of several SoA rule-based approaches as being from different underlying OOD distributions and different regularization (see next section for a more detailed discussion).

In summary, our contributions are as follows: **1.** We introduce a novel theory and framework for deriving OOD features from neural networks. This involves the formulation of OOD features as a variational problem that formulates OOD features as *random features* through a novel loss functional that contains two terms, one that maximizes the *KL divergence* between the random feature under ID and OOD distributions and another term, the *Information Bottleneck*, which extracts the information from the data that is relevant for OOD detection. **2.** We develop the techniques to optimize the loss functional using the calculus of variations, and specifically derive a computationally feasible algorithm in the one-dimensional data case. **3.** Using our framework, we show how the OOD shaping functions change based on various data distributions. We relate the mean value of our OOD features to existing OOD shaping functions. **4.** We introduce a novel piece-wise linear OOD feature shaping function predicted through our theory, and show that it leads to state-of-the-art results on OOD benchmarks.

## 1.1. Related Work

We briefly review related work; the reader is referred to (Yang et al., 2022) for a survey. Post-hoc approaches of OOD detection, which are applied to pre-trained models without additional training, have focused on constructing scoring functions to differentiate OOD from in-distribution data, leveraging confidence scores (Hendrycks & Gimpel, 2018a; Zhang & Xiang, 2023; Liang et al., 2020), energy-based metrics (Liu et al., 2021; Wu et al., 2023; Elflein et al., 2021) and distance-based measures (Lee et al., 2018; Sun et al., 2022). For example, MSP (Hendrycks & Gimpel, 2018a) used the maximum softmax probability as a confidence score. ODIN (Liang et al., 2020) improved OOD detection by applying temperature scaling and adding small perturbations to input data before computing the maximum

softmax probability. (Ren et al., 2019) proposes to use the likelihood ratio, which has been proposed over likelihoods, which do not work well (Kirichenko et al., 2020). (Lee et al., 2018) leveraged Mahalnobis distance to compute the distance between features and classes. KNN (Sun et al., 2022) uses a non-parametric approach. Energy methods (Liu et al., 2021) present an alternative to softmax scores by employing the Helmholtz free energy. Energy scoring has been adopted by several *OOD feature-shaping* approaches; feature-shaping is the focus of our work.

**Feature-shaping approaches to OOD detection**: Several methods perform OOD detection by computing features of the output of layers of the neural network (Sun et al., 2021; Kong & Li, 2023; Djurisic et al., 2022; Fort et al., 2021b; Zhao et al., 2024) before being input to a score. In ReAct (Sun et al., 2021), the penultimate layer outputs are processed element-wise by clipping large values. It is empirically noted that OOD data results in large spikes in activations, which are clipped to better separate the ID and OOD distributions. BFAct (Kong & Li, 2023) uses the Butterworth filter to smoothly approximate the clipping. ASH computes features by sparsifying intermediate outputs of the network by flooring small values to zero and passing larger values with possible scaling. DICE (Sun & Li, 2022) is another approach to sparsification. Different than element-wise approaches, ASH then does vector processing of the shaped feature before input to a score. VRA (Xu et al., 2023) and (Zhang et al.) derive element-wise shaping functions by an optimization approach.

**Optimization-based approaches for feature shaping**: (Xu et al., 2023) formulates a loss function for deterministic OOD features that aims to separate the means of ID and OOD feature distributions, and regularization is added to keep the OOD feature near the identity through the L2 norm. (Zhao et al., 2024) analyzes a similar loss function but with point-wise rather than L2 regularization. They further offer simplifications to remove the reliance on the OOD distribution. These works have introduced the novel idea of formulating OOD features through a loss function. Our approach offers several advantages. Over (Zhao et al., 2024), we present a framework in which we can study the OOD feature as a function of the underlying OOD distribution. This shows the implicit assumptions in several existing methods. In contrast, (Zhao et al., 2024) aims to remove dependence on the OOD distribution. Our results show that feature shaping can vary as a function of the underlying OOD distribution. Over (Zhao et al., 2024; Xu et al., 2023), our theory offers an explanation of qualitative properties of existing SoA methods. For instance, clipping of large values in OOD features (of ReAct (Sun et al., 2021)) is associated with a higher Information Bottleneck (IB) regularization which is needed for noisier OOD datasets. Negative slope at large values in (Zhao et al., 2024; Xu et al., 2023) is associated

with low IB regularization. Also, pruning of small feature values in (Xu et al., 2023; Djurisic et al., 2022) is associated with OOD distributions with heavier tails. See Section 4 for more technical details.

## 2. Variational Formulation of OOD Features

We formulate OOD features as an optimization problem. For the sake of the derivation, we will assume that the probability distributions of ID and OOD features from the network are given in this section. In practice, the ID can be estimated by training data. In Section 4, we will then study the OOD features under various distributions to show how features vary with distribution and offer plausible assumptions made by existing feature shaping approaches. We will also make reasonable assumptions on the OOD distribution to derive new prescriptive OOD features for use in practice.

Current OOD features in the OOD literature are computed by processing features from the neural network through a deterministic function (e.g., clipping). In contrast, we propose to generalize that approach by allowing for *random* functions. Let $Z$ denote the feature (a random variable) from the network (penultimate or intermediate layer feature). We denote by $\tilde{Z}$ the random OOD feature (a random variable) that we seek to determine. The distribution of $\tilde{Z}$ is denoted $p(\tilde{z}|z)$. Thus, rather than solving for a deterministic function $f(Z)$, we instead solve for a random feature $\tilde{Z}$ represented through $p(\tilde{z}|z)$ as in Information Theory (Cover, 1999). Thus, given a feature $z$, the OOD feature is $\tilde{Z} \sim p(\tilde{z}|z)$. We will primarily be concerned with the mean value of the distribution in this paper to relate to other feature shaping methods. Let $X$ be the random variable indicating the data (e.g., image, text), and $Y$ be the random variable indicating in- ($Y = 0$) and out-of- ($Y = 1$) distribution data. Note this forms a Markov Chain $Y \to X \to Z \to \tilde{Z}$. The Markov Chain property is needed to construct one of the terms of our loss function, discussed next.

We propose a novel loss functional to design the OOD random feature. This loss functional is defined on $p(\tilde{z}|z)$. The first term aims to separate the ID and OOD distributions of the random feature $\tilde{Z}$. This is natural since we would like to use the OOD feature to separate the data into in or out-of-distribution. To achieve this separation, we propose to maximize the symmetrized KL-divergence between $p(\tilde{z}|Y = 0)$ and $p(\tilde{z}|Y = 1)$. Note recent work (Zhao et al., 2024) also seeks to separate distributions, however, differently than our approach as only the means of the distribution are separated. Also, note that $p(\tilde{z}|Y = y)$ is a function of $p(\tilde{z}|z)$, the variable of optimization, and thus the KL term is a function of $p(\tilde{z}|z)$. This term is defined as follows:

$$D_{KL}(p(\tilde{z}|z)) = D_{KL}[p(\tilde{z}|Y = 1) \,\|\, p(\tilde{z}|Y = 0)] + \\ D_{KL}[p(\tilde{z}|Y = 0) \,\|\, p(\tilde{z}|Y = 1)], \quad (1)$$

where

$$D_{KL}[p \,\|\, q] = \int p(x) \log \frac{p(x)}{q(x)} \, \mathrm{d}x, \quad \text{and} \quad (2)$$

$$p(\tilde{z}|y) = \int p(\tilde{z}|z)p(z|y) \, \mathrm{d}z. \quad (3)$$

Note that we have used that $p(\tilde{z}|z, y) = p(\tilde{z}|z)$ as the feature is constructed the same for both ID and OOD data. These equations shows the dependence of the OOD feature distributions on $p(\tilde{z}|z)$. The KL divergence is a natural choice for separating distributions and a standard information-theoretic quantity.

Unconstrained maximization of KL divergence is ill-posed, and regularization is needed. Also, it is possible to reduce the dimensions of $Z$ to a few dimensions that are maximally separated but remove information necessary to fully characterize OOD data. Therefore, we need to ensure that $\tilde{Z}$ contains all the information relevant to accurately determine OOD data. With these considerations, we aim to compress the dimensions of $Z$ to form a simple/compact feature, but in a way that preserves the OOD information (contained in the variable $Y$). To achieve this, we adapt the Information Bottleneck (Tishby et al., 2000). In the Information Bottleneck method, the quantization of a random variable $X$ is considered to form the random variable $T$ in such a way to preserve information about a random variable $Y$, where $Y$ forms a Markov Chain with $X$. A functional is formulated such that when minimized forms $T$. This is precisely the functional we would like to determine $\tilde{Z}$ (where $\tilde{Z}$ is analogous to $T$ and $Z$ is analogous to $X$). The second term of our functional, following from (Tishby et al., 2000), is

$$\mathrm{IB}(p(\tilde{z}|z)) = I(Z; \tilde{Z}) - \beta I(\tilde{Z}; Y), \quad (4)$$

where $I$ indicates mutual information, and $\beta > 0$ is a hyperparameter. The first term of (4) is the compression term that measures the mutual information between $Z$ and $\tilde{Z}$; this term is minimized and thus the term favors $\tilde{Z}$ to be a compressed version of $Z$. The second term maximizes the mutual information between $\tilde{Z}$ and $Y$, and thus favors $\tilde{Z}$ to retain OOD relevant information.

Thus, our combined loss functional is

$$L(p(\tilde{z}|z)) = -D_{KL}(p(\tilde{z}|z)) + \alpha \mathrm{IB}(p(\tilde{z}|z)), \quad (5)$$

which is minimized to determine the conditional distribution of $\tilde{Z}$, $p(\tilde{z}|z)$, and $\alpha > 0$ is a hyperparameter. Our goal is to determine the optimal $p(\tilde{z}|z)$, which can then be used with a score function to determine whether data $z$ is OOD or not. Note that we are seeking to optimize over the set of continuous probability distributions, which forms an infinite dimensional optimization problem. To gain intuition into the loss functional above, in particular to see that it forms a well-posed problem and that IB regularization is needed, we

analyze a simple case with 1D Gaussian distributions that result in closed form solution in Appendix A. We verify in the next section that the loss functional, for more complex distributions/features, yields well-posed problems and hence result in an optimal solution.

## 3. Optimization for OOD Features

In this section, we discuss the optimization of the loss functional (5). The loss functional is defined on continuous probability density functions $p(\tilde{z}|z)$, where $z, \tilde{z}$ are continuous. This is an infinite dimensional optimization problem, and to find the optimal feature one can use the calculus of variations to determine the gradient of $L$ (Troutman, 2012). Setting the gradient to zero and solving for the probability distribution that satisfies the equation gives the necessary conditions for the optimizer. For our loss, that does not yield a closed form solution and so we instead use the gradient to perform a gradient descent.

### 3.1. Loss Under Element-wise Independence of Feature

Because formulating numerical optimization for general multi-dimensional distributions is difficult, we make some simplifications to gain insights to our theory and approach. Even with these simplifications, we will show that the approach can explain popular approaches in the literature and lead to a new state of the art approach. Our first simplification (which is similar to element-wise processing assumptions made in existing methods, e.g., (Sun et al., 2021; Zhao et al., 2024)) is to assume that the conditional feature distribution $p(\tilde{z}|z)$ can be factorized as $p(\tilde{z}|z) = \prod_{i=1}^{n} p(\tilde{z}_i|z)$, which assumes conditional independence of the components of $\tilde{z}$ and that each component has the same conditional distribution. We also assume that $p(z|y) = \prod_{i=1}^{n} p(z_i|y)$, that is, the components of $z$ are independent conditioned on $y$. Under these assumptions, the optimization of the loss functional (5) reduces to the optimization of several optimization problems defined on one-dimensional probability distributions from each feature component (see Appendix B for details):

$$\underset{p(\tilde{z}_i|z_i)}{\arg\min} L_i(p(\tilde{z}_i|z_i)), \quad i \in \{1, \ldots, n\}, \quad (6)$$

where

$$L_i(p(\tilde{z}_i|z_i)) = -D_{KL}[p(\tilde{z}_i|0) \,||\, p(\tilde{z}_i|1)] - $$
$$D_{KL}[p(\tilde{z}_i|1) \,||\, p(\tilde{z}_i|0)] + \alpha[I(\tilde{Z}_i; Z_i) - \beta I(\tilde{Z}_i; Y)]. \quad (7)$$

Thus, we next provide an optimization procedure for the loss functionals above, defined on one-dimensional distributions. For simplicity of notation, we now omit the $i$ subscripts.

### 3.2. Gradient of Loss Functional

We will use gradient descent to optimize the loss functional. Since the problem is non-convex, gradient descent is a natural choice. Given the infinite dimensional problem, we use the calculus of variations to compute the gradient.

We perform the computation for the gradient of (5) in Appendix C and summarize the result in the following theorem:

**Theorem 3.1** (Gradient of Loss). *The gradient of $D_{KL}(p(\tilde{z}|z)))$ (1) with respect to $p(\tilde{z}|z)$ is given (up to an additive function of z) by*

$$\nabla_{p(\tilde{z}|z)} D_{KL} = p(z|0) \cdot [l(z) \log l(\tilde{z}) - l(\tilde{z})] $$
$$- p(z|1) \cdot \left[ l(z)^{-1} \log l(\tilde{z}) + l(\tilde{z})^{-1} \right], \quad (8)$$

*where $p(z|y) = p(z|Y = y)$, $p(\tilde{z}|y) = p(\tilde{z}|Y = y)$ and*

$$l(z) = \frac{p(z|1)}{p(z|0)}, \quad and \quad l(\tilde{z}) = \frac{p(\tilde{z}|1)}{p(\tilde{z}|0)}. \quad (9)$$

*The gradient of $IB(p(\tilde{z}|z))$ (4) is given by $\nabla_{p(\tilde{z}|z)} IB =$*

$$\sum_{y \in \{0,1\}} p(y)p(z|y) \left[ \log \frac{p(\tilde{z}|z)}{p(\tilde{z})} - \beta \log \frac{p(\tilde{z}|y)}{p(\tilde{z})} \right]. \quad (10)$$

*The gradient of the full loss $L$ in (5) is then*

$$\nabla_{p(\tilde{z}|z)} L = -\nabla_{p(\tilde{z}|z)} D_{KL} + \alpha \nabla_{p(\tilde{z}|z)} IB. \quad (11)$$

To simplify further and study a model that more closely resembles OOD feature shaping functions in the literature, we make the following assumption:

$$p(\tilde{z}|z) \sim \mathcal{N}(\mu(z), \sigma_c(z)), \quad (12)$$

where $\mathcal{N}$ indicates Gaussian distribution, $\tilde{z}, z \in \mathbb{R}$ and $\mu, \sigma_c : \mathbb{R} \to \mathbb{R}$ are the mean/standard deviation. We use the sub-script c to denote "conditional" to distinguish it from other sigmas used below. We can think of this model as random perturbations of a deterministic feature shaping function $\mu$. The OOD's feature mean value for a given network feature $z$ is $\mu(z)$. The closer $\sigma_c$ is to zero, the closer the approach is to deterministic feature shaping. Note if the optimization turns out to result in $\sigma_c = 0$, then deterministic functions would be optimal. In our numerous simulations, this does not happen and thus random OOD features appear to be more optimal. We now compute the gradients with respect to $\mu$ and $\sigma_c$:

**Theorem 3.2** (Loss Gradient Under Gaussian Random OOD Feature (12)). *The gradient of the loss (5) under (12) is*

$$\nabla_\mu L(z) = \int \frac{\nabla_{p(\tilde{z}|z)} L(\tilde{z}, z)}{\sigma_c^2(z)} [\tilde{z} - \mu(z)] p(\tilde{z}|z) \, d\tilde{z} \quad (13)$$

$$\nabla_{\sigma_c} L(z) = \int \frac{\nabla_{p(\tilde{z}|z)} L(\tilde{z}, z)}{\sigma_c(z)} \left[ \frac{(\tilde{z} - \mu(z))^2}{\sigma_c^2(z)} - 1 \right] p(\tilde{z}|z) \, d\tilde{z}, \quad (14)$$

*where $\nabla_{p(\tilde{z}|z)} L$ is given in (11).*

## 3.3. Numerical Optimization of Loss

We implement a gradient descent algorithm using a discretization of the continuum equations above. We choose a uniform discretization of the space of $z$, i.e., $\{z_i\}_i \subset \mathbb{R}$. We represent $\mu$ and $\sigma_c$ through their samples: $\mu_i = \mu(z_i)$ and $\sigma_{c,i} = \sigma_c(z_i)$. We specify formulas for $p(\tilde{z})$ and $p(\tilde{z}|y)$ under the discretization, which will be required in the computation of the approximation to the gradient:

$$p(\tilde{z}|y) = \sum_i p(\tilde{z}|z_i)p(z_i|y)\Delta z_i$$

$$= \sum_i \frac{1}{\sigma_{c,i}} G_{\sigma_{c,i}}(\tilde{z} - \mu_i)p(z_i|y)\Delta z_i \quad (15)$$

$$p(\tilde{z}) = \sum_y p(y)p(\tilde{z}|y). \quad (16)$$

Thus, $p(\tilde{z}|y)$ is approximated as a mixture of Gaussians. The gradient descent is shown in Algorithm 1, which assumes ID and OOD distributions and determines the Gaussian random feature parameterized through $\mu$ and $\sigma_c$.

The complexity for this optimization (which is done off-line in training) is $\mathcal{O}(NMK)$ where $N$ is the number of samples of $p(z|y)$, $M$ is the samples of $p(\tilde{z}|z)$ and $K$ is the number of gradient descent iterations. On a single A100 GPU, this took less than a minute.

## 4. A Study of OOD Features vs Distribution

In this section, we study the resulting OOD features based on various choices of distributions using the algorithm in the previous section, and relate these choices to OOD feature shaping techniques that are present in the literature. Note that while in practice the OOD distribution is unknown, our theory nevertheless suggests the underlying distributional assumptions of existing methods. This is useful to understand when these methods will generalize as a function of the type of OOD data. We will also derive a generic OOD shaping function, encompassing properties of several distributions, and show that this shaping function can lead to SoA performance in the next section. Note in practice, we have observed that distributions from OOD datasets to exhibit similarities to the distributions studied, see Appendix H. We will further rationale on studying these distributions below.

For this study, we will assume the assumptions of Section 3.3, i.e., that the OOD features are element-wise independent and that the OOD feature is Gaussian, i.e., $p(\tilde{z}|z) \sim \mathcal{N}(\mu(z), \sigma_c(z))$. We will further assume that the ID distribution is Gaussian, i.e., $p(z|0) \sim \mathcal{N}(\mu_0, \sigma)$. We make this assumption for simplicity and that features in network layers can be approximated well with a Gaussian, as evidenced empirically in (Xu et al., 2023). We will study three OOD distributions next: Gaussian, Laplacian and a distribution we propose based on the Inverse Gaussian.

---

**Algorithm 1** 1D Gaussian Random Feature Computation

**Input:** IN/OOD Distributions $p(z|y)$, $\alpha, \beta$ and learning rate $\eta$
**Output:** Converged mean $\mu_i$, std $\sigma_{c,i}$ for each $i$
Initialize: $\mu_i = z_i$, $\sigma_{c,i} = \text{const}$
**for** $n$ iterations **do**
    **for** $z_i$ **do**
        Compute a discretization of $\tilde{z}$ in its likely range: $\tilde{z}_i^j \in (\mu_i - k\sigma_{c,i}, \mu_i + k\sigma_{c,i})$ where $k \geq 3$
        **for** $\tilde{z}_j^i$ **do**
            Compute $\nabla_{p(\tilde{z}|z)} L(\tilde{z}_j^i, z_i) =$

$$p(z_i|0) \cdot \left[ l(z_i) \log l(\tilde{z}_j^i) - l(\tilde{z}_j^i) \right] -$$

$$p(z_i|1) \cdot \left[ l(z_i)^{-1} \log l(\tilde{z}_j^i) + l(\tilde{z}_j^i)^{-1} \right] +$$

$$\alpha \sum_{y \in \{0,1\}} p(y)p(z_i|y) \left[ \log \frac{p(\tilde{z}_j^i|z_i)}{p(\tilde{z}_j^i)} - \beta \log \frac{p(\tilde{z}_j^i|y)}{p(\tilde{z}_j^i)} \right]$$

        **end for**
        Compute $\nabla_\mu L(z_i) =$

$$\sum_j \frac{\nabla_{p(\tilde{z}|z)} L(\tilde{z}_j^i, z_i)}{\sigma_{c,i}^2} (\tilde{z}_j^i - \mu_i) p(\tilde{z}_j^i|z_i)\Delta z_i$$

        Compute $\nabla_{\sigma_c} L(z_i) =$

$$\sum_j \frac{\nabla_{p(\tilde{z}|z)} L(\tilde{z}_j^i, z_i)}{\sigma_{c,i}} \left[ \frac{(\tilde{z}_j^i - \mu_i)^2}{\sigma_{c,i}^2} - 1 \right] p(\tilde{z}_j^i|z_i)\Delta z_i$$

    **end for**
    **for** $z_i$ **do**

$$\mu_i \leftarrow \mu_i - \eta \nabla_\mu L(z_i)$$
$$\sigma_{c,i} \leftarrow \sigma_{c,i} - \eta \nabla_{\sigma_c} L(z_i)$$

    **end for**
**end for**

---

**Gaussian OOD**: First, we study the case of a Gaussian for the OOD distribution, as its the most common distribution in probabilistic analysis. Let $p(z|1) \sim \mathcal{N}(\mu_1, \sigma)$. For illustration, we choose $\mu_0 = -0.5, \mu_1 = 0.5$ and $\sigma = 0.5$ and $\alpha = 1.0, \beta = 10$. The resulting converged result of the optimization for $\mu$ and $\sigma_c$ is shown in Figure 1 (positive part). No feature shaping would mean that $\mu$ is the identity map, and $\sigma_c = 0$; this solution is plotted in dashed blue. Notice that the optimal mean value is not the identity. The mean indicates that the feature has positive slope for small values of $|z|$ (similar to (Sun et al., 2021)) and negative slope for large values of $|z|$ (similar to (Zhao et al., 2024)). In Appendix E.2, we show that under different distribution parameters, one can get negative values for small $|z|$ as in (Zhao et al., 2024). Interestingly, the optimal standard deviation $\sigma_c(z)$ is non-zero, indicating randomness in this

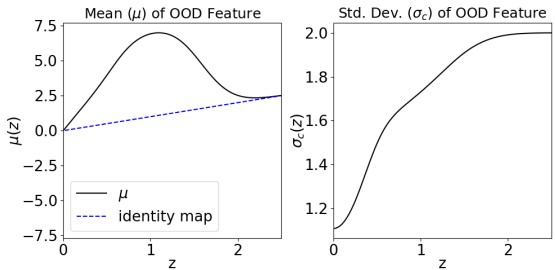

*Figure 1.* OOD Gaussian Feature Under Gaussian ID/OOD Distributions. Mean (left), standard deviation (right) of the feature.

case is beneficial in terms of the loss. In fact, in all of our simulations across distributions and their hyperparameters, we've observed non-zero standard deviation.

In Figure 2(a), we show the effects of the Information Bottleneck weight $\alpha$. The impact of $\beta$ on the shape is studied in Appendix E.1. For $\alpha$ larger (higher regularization), the mean of the feature becomes flat for $|z|$ large, similar to clipping that is used in popular methods (Sun et al., 2021; Xu et al., 2023).[1] See Figure 3 for a plot of existing feature shaping methods. Even under the simplifying Gaussian assumptions, we see that the our shaping functions have properties of existing methods.

**Laplacian OOD Distribution**: Next, we consider the Laplacian distribution for the OOD distribution, i.e., $p(z|1) = \frac{1}{2b}\exp\left(-|z - \mu_1|/b\right)$. The intuition for choosing this distribution is that it has a heavier tail than the Gaussian, and thus, is better able to model outliers, and it seems reasonable OOD data would be considered outliers. We show the result of the mean of the feature in Figure 2(b). We notice that when $|z|$ is small, the mean OOD feature is zero, which indicates a suppression of low values (this is used in VRA (Xu et al., 2023) and ASH (Djurisic et al., 2022)). Note that this is consistent across $\alpha$ values, larger values increases the suppression region. We also see that large values of $|z|$ are being clipped or suppressed (with larger $\alpha$) approaching a zero slope. The jump discontinuity is also present in VRA and ASH. There also appears to be a positively sloped linear function for intermediate values of $|z|$, similar to VRA.

**Inverse Gaussian OOD Distribution**: Next, we consider a distribution that may be a distribution that generically holds for OOD data and can be used in the absence of prior information of the OOD distribution. If the ID distribution is Gaussian, we can formulate a distribution that has high probability outside the domain that the ID distribution has high probability. To this end, one can consider a variant of the Inverse Gaussian defined as follows. Let $d(z) = |z - \mu_0|/\sigma$ where $\mu_0, \sigma$ are the mean and standard

---

[1]Note our approach does not assume energy scoring like SoA, which changes the scale of the features. However, our approach uncovers and explains similar shapes irrespective of the score.

deviation of the ID distribution. This is a distance to the ID distribution. We would like the OOD distribution to have high probability when $d(z)$ is large, and thus we consider $p(z|1) \sim IG(d(z); \mu_1, \lambda)$ where IG denotes the inverse Gaussian distribution:

$$p_{IG}(x; \mu, \lambda) = \sqrt{\frac{\lambda}{2\pi x^3}} \exp\left(-\frac{\lambda(x - \mu)^2}{2\mu^2 x}\right), \quad (17)$$

which is plotted in Appendix D. Note that there is some overlap of this distribution with the ID Gaussian. As noted in Figure 2(c), the Inverse Gaussian distribution results in a qualitatively similar OOD feature compared to the Laplacian distribution: suppression of small $|z|$ values and clipping/flattening of large $|z|$ values and a positively sloped linear function for intermediate values of $|z|$. For $\alpha$ large we have flattening similar to clipping and $\alpha$ smaller results in a negative slope similar to the other distributions.

We summarize key observations. Clipping as done in Re-Act seems to be a universal property across all the OOD distributions for large regularization. In the next section we show that for noiser OOD datasets larger regularization is beneficial, and so the clipping mitigates noise, which is noted in (Sun et al., 2021). Next, the OOD distributions that are heavier tailed result in suppression (zeroing out) of low $|z|$ values. This is consistent with the VRA and ASH methods. All distributions yield a positively sloped region for intermediate values of $|z|$. Our results suggest ReAct and FS-OPT may be operating under an implicit Gaussian OOD assumption for high regularization (ReAct) and low regularization (FS-OPT). VRA and ASH seem to implicitly assume heavier tailed OOD distributions.

**Piecewise Linear Shaping**: The above mean shaping functions (from Gaussian, Laplace and Inverse Gaussian OOD distributions) all approximately fit in a particular piecewise linear function family as shown in Figure 4, where $z_1, z_2, y_0, y_{1a}, y_{1b}, m_1, m_2$ are hyperparameters. Therefore, in practice, if the distribution is unknown, one can choose this family of shaping functions that would implicitly assume any of the aforementioned three distributions. Because many existing SOA methods implicitly make one of the three distributional assumptions, this family makes more general distributional assumptions than existing SOA, thus potentially offering generalization to more OOD datasets while not being too general so as to lose discriminability. In the experimental section we explore this family of shaping functions, and show we can obtain SoA results.

## 5. Implementation of New OOD Detection

In this section, we provide the implementation details for our new approaches to OOD detection, using the simplifying assumptions presented in Section 3. We provide the details for two cases where the ID/OOD distributions are known

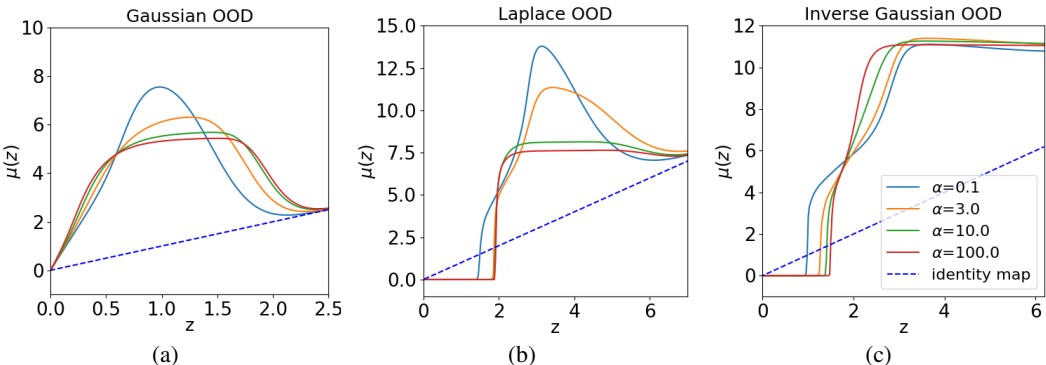

*Figure 2.* The mean of the OOD Gaussian Feature under the Gaussian (left), Laplace (middle) and Inverse Gaussian (right) OOD distributions for varying weights on the Information Bottleneck, $\alpha$. For all plots, $\beta = 10$. For the Gaussian case, $p(z|0) \sim \mathcal{N}(-0.5, 0.5)$ and $p(z|1) \sim \mathcal{N}(0.5, 0.5)$. For the Laplace case, $p(z|0) \sim \mathcal{N}(0, 0.66)$ and $p(z|1) \sim Lap(0, 1)$. In the Inverse Gaussian case, $p(z|0) \sim \mathcal{N}(0, 0.66)$ and $p(z|1) \sim IG(d(z); 3.3, 15)$. For visualization purpose, we only show the positive part.

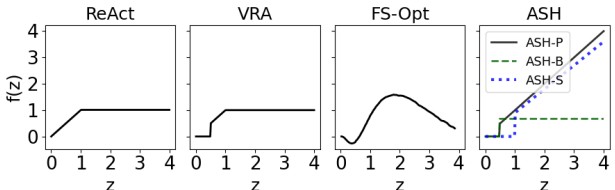

*Figure 3.* Plot of existing feature shaping functions from SoA methods: ReAct (Sun et al., 2021), VRA (Xu et al., 2023), FS-Opt (Zhao et al., 2024), and variants of ASH (Djurisic et al., 2022).

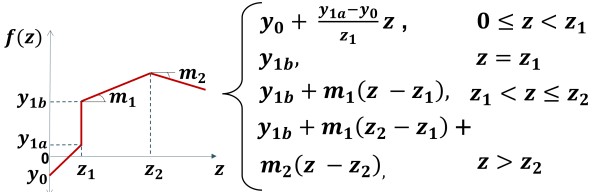

*Figure 4.* A piece-wise linear family of functions that approximately encompasses the mean value of our OOD feature shaping functions across OOD distributions examined in this paper.

and unknown. In the latter case, we apply the piecewise family of feature shaping derived in the previous section (Figure 4). We assume that a validation set of ID and OOD data is available (as in existing literature) and the choices are given in our experiments section. A trained neural network is also provided. The network feature vectors and their ID/OOD label for the validation set are $\{z_i, y_i\}$. Consistent with our simplifying assumptions and literature, each component of the network feature $z$ is processed independently and for this paper, they will be processed by the same shaping function $\mu$.

**Off-line-Training**: Under the case that the forms of the ID and OOD distributions are known, the hyperparameters of the distributions are estimated from the validation set (rasterizing the vector data). Using the fitted distributions, Algorithm 1 is run to compute the optimal $\mu^*, \sigma_c^*$. In the case that the distributions are unknown, we assume that

the feature shape fits the piecewise family in the previous section (i.e., the OOD distributions are one of Gaussian, Laplacian or IG). The hyper-parameters for the piecewise family are tuned by e.g., minimizing the false positive rate at true positive rate of 95% (FPR95) metric on the validation set - this gives the optimal shaping function $\mu^*$.

**Online Operation**: During operation, the network feature $z$ is extracted, and then shaped via the function $\tilde{z} = \mu^*(z) = (\mu^*(z_1), \ldots, \mu^*(z_n))$. Subsequently, $\tilde{z}$ is input to a scoring function (e.g., in this paper, energy score (Liu et al., 2021)), which is then thresholded to produce the ID/OOD label.

## 6. Experiments

We validate our theory by comparing our new shaping function to SoA for OOD detection on standard benchmarks.

**Datasets and Model architectures.** We experiment with ResNet-50(He et al., 2016), MobileNet-v2 (Sandler et al., 2018), vision transformers ViT-B-16 and ViT-L-16 (Dosovitskiy et al., 2021) with ImageNet-1k (Russakovsky et al., 2015) as ID data, and benchmark on the OOD datasets/methods used in (Zhao et al., 2024). For the ImageNet benchmark, we evaluate performance across eight OOD datasets: Species (Hendrycks et al., 2022), iNaturalist (Horn et al., 2018), SUN (Xiao et al., 2010), Places (Zhou et al., 2018), OpenImage-O (Wang et al., 2022), ImageNet-O (Hendrycks et al., 2021), Texture (Cimpoi et al., 2014), and MNIST (Deng, 2012). Moreover, we also experiment with CIFAR 10 and CIFAR 100 as ID data, for which we use a ViT-B-16 (Dosovitskiy et al., 2021) finetuned on CIFAR10/100 consistent with (Fort et al., 2021a), and a MLP-Mixer-Nano model trained on CIFAR10/100 from scratch. We evaluate eight OOD datasets: TinyImageNet (Torralba et al., 2008), SVHN (Netzer et al., 2011), Texture (Cimpoi et al., 2014), Places365 (Zhou et al., 2018), LSUN-Cropped (Yu et al., 2016), LSUN-Resized (Yu et al., 2016), iSUN

(Xu et al., 2015), and CIFAR100/ CIFAR10 (CIFAR 100 treated as OOD for CIFAR 10, and vice-versa).

We compare our results against the SoA methods across two categories - penultimate layer element-wise feature shaping approaches, which our theory currently applies to, and other methods. Penultimate layer feature shaping approaches involve element-wise feature shaping functions applied directly to the penultimate layer of the model before computing the energy score for OOD detection. Approaches in this category are: Energy (Liu et al., 2021), ReAct (Sun et al., 2021), BFAct (Kong & Li, 2023), VRA-P (Xu et al., 2023) and FS-OPT (Zhao et al., 2024). The second category, which are not directly comparable to our approach because they may not involve feature shaping or additions to feature matching, but are included for reference, are softmax-based confidence scoring (MSP (Hendrycks & Gimpel, 2018b)), input perturbation and temperature scaling (ODIN (Liang et al., 2020)), intermediate-layer shaping and subsequent processing by following layers (ASH-P, ASH-B, ASH-S (Djurisic et al., 2022)), or weight sparsification (DICE (Sun & Li, 2022)).

As in ReAct (Sun et al., 2021), for ImageNet-1k benchmarks we use a validation set comprising the validation split of ImageNet-1k as ID data, and Gaussian noise images as OOD data, generated by sampling from $\mathcal{N}(0,1)$ for each pixel location, to tune the hyperparameters of our piecewise linear activation shaping function. For CIFAR 10/100 benchmarks, following ODIN (Liang et al., 2020), we employ a random subset of the iSUN dataset (Xu et al., 2015) as validation OOD data for our hyperparameter tuning. As ID validation data for CIFAR10/100 we use the test splits of the corresponding datasets. The hyperparameters are optimized using Bayesian optimization (Frazier, 2018), by minimizing the FPR95 metric on the validation set. Resulting hyperparameters are reported in Appendix G.

**Metrics.** We utilize two standard evaluation metrics, following (Sun et al., 2021; Zhao et al., 2024): FPR95 - the false positive rate when the true positive rate is 95% (abbreviated as FP), and the area under the ROC curve (AU).

**Results.** The results on the ImageNet-1k benchmarks (Table 1) and the CIFAR 10/100 benchmarks (Table 2) demonstrate that our approach achieves state-of-the-art performance among comparable feature-shaping methods in the previously mentioned category of methods. Specifically, when compared to pointwise feature-shaping techniques such as ReAct, BFAct, VRA-P, and FS-OPT, our method consistently outperforms these approaches, yielding the best overall results in this category.

While ASH variants marginally outperform our method in some cases, it is important to note that ASH employs a fundamentally different approach. ASH modifies activations

through intermediate layer pruning and rescaling of features, which are then routed back into the network for further processing, and thereby is not an element-wise feature shaping approach. Our theory currently does not address this case.

For the vision transformers ViT-B-16 and ViT-L-16, our method achieves the lowest FP among all competing methods, providing evidence of generalization across different architectures. Overall, our results demonstrate that our feature shaping is highly competitive with the latest SoA, along with providing a theoretical explanation.

**Computational Time**: The inference cost of our feature shaping method is on the order of microseconds for a 256×256×3 image, using PyTorch on an NVIDIA A100-80GB GPU. This is comparable to other piecewise linear shaping approaches such as ReAct and VRA.

**Regularization as a Function of OOD Data.** We study how IB regularization should be chosen with respect to properties of OOD data. This is important in practical scenarios. In particular, we conduct an experiment to suggest higher IB regularization is beneficial for "noisier" OOD datasets.

To this end, we conduct a series of controlled experiments using ResNet-50 trained on the ImageNet-1k dataset. We aim to determine the structure of optimal feature-shaping functions as a function of noise. We apply additive Gaussian noise $\mathcal{N}(0,\sigma)$ to the ImageNet-1k validation set and consider them as OOD data. The standard deviation $\sigma$ used are $\{25, 50, 100, 150, 255\}$ to create 5 OOD datasets. Visualizations of this data and activation patterns are shown in Appendix F. We observe that this data closely resembles the high variance of activation patterns in OOD datasets as noted in (Sun et al., 2021), and so our noisy data serves to mimic OOD data with varying noise levels. By examining how the learned features adapt under different noise levels, we gain insights into the relationship between the OOD data and the IB regularization term for optimal shaping.

In Figure 5, we plot the IB term of the obtained optimal shaping function optimized over hyperparameters at each noise level. Note we have used the Laplacian OOD distribution to estimate the IB term (Inverse Gaussian also results in similar results). It is seen that higher noise levels result in optimal shaping functions with lower IB values, which correspond to higher degree of regularization of the IB term in our loss functional. Thus, noisier OOD datasets require higher IB regularization for best OOD detection performance.

## 7. Conclusion and Future Work

We have presented a novel theory for OOD feature construction. OOD features were computed as the optimizer of a loss functional consisting of a term maximizing the KL distance between resulting features under ID and OOD distributions

|  | Method | ResNet50 FP↓ | ResNet50 AU↑ | MobileNetV2 FP↓ | MobileNetV2 AU↑ | ViT-B-16 FP↓ | ViT-B-16 AU↑ | ViT-L-16 FP↓ | ViT-L-16 AU↑ |
|---|---|---|---|---|---|---|---|---|---|
| Element-wise Feature Shaping | Energy | 60.97 | 81.01 | 61.40 | 82.83 | 73.96 | 67.65 | 74.89 | 70.11 |
| | ReAct | 42.29 | 86.54 | 54.19 | 85.37 | 73.82 | 76.86 | 76.16 | 81.07 |
| | BFAct | 43.87 | 86.01 | 52.87 | 85.78 | 77.64 | 80.16 | 84.02 | 81.12 |
| | VRA-P | 37.97 | 88.58 | 49.98 | 86.83 | 98.39 | 35.66 | 99.58 | 16.70 |
| | FS-OPT | 39.75 | 88.56 | 51.77 | 86.62 | 69.52 | **82.66** | 72.17 | 83.23 |
| | Ours | **35.82** | **89.36** | **46.97** | **87.49** | 67.73 | 81.06 | **66.67** | **83.92** |
| Other methods | MSP | 69.30 | 76.26 | 72.58 | 77.41 | 69.84 | 77.40 | 70.59 | 78.40 |
| | ODIN | 61.56 | 80.92 | 62.91 | 82.64 | 69.25 | 72.60 | 70.35 | 74.51 |
| | ASH-P | 55.30 | 83.00 | 59.41 | 83.84 | 99.36 | 21.17 | 99.18 | 20.27 |
| | ASH-B | 35.97 | 88.62 | 43.59 | 88.28 | 94.87 | 46.68 | 93.72 | 38.95 |
| | ASH-S | 34.70 | 90.25 | 43.84 | 88.24 | 99.48 | 18.52 | 99.42 | 18.61 |
| | DICE | 45.32 | 83.64 | 49.33 | 84.63 | 89.68 | 71.32 | 72.38 | 67.08 |

*Table 1.* ImageNet-1k Benchmarking Results. Lower FP is better, higher AU is better. Bold values highlight the best results among element-wise feature shaping methods. Underlined values indicate best results among other methods.

|  | Method | CIFAR10 DenseNet101 FP↓ | CIFAR10 DenseNet101 AU↑ | CIFAR10 MLP-N FP↓ | CIFAR10 MLP-N AU↑ | CIFAR100 DenseNet101 FP↓ | CIFAR100 DenseNet101 AU↑ | CIFAR100 MLP-N FP↓ | CIFAR100 MLP-N AU↑ |
|---|---|---|---|---|---|---|---|---|---|
| Element-wise Feature Shaping | Energy | 31.72 | 93.51 | 63.95 | **82.84** | 70.80 | 80.22 | 79.90 | 75.30 |
| | ReAct | 82.00 | 76.46 | 64.34 | 81.85 | 77.00 | 78.30 | 79.99 | 75.87 |
| | BFact | 84.40 | 74.39 | 78.02 | 72.68 | 80.27 | 73.36 | 80.05 | **76.58** |
| | VRA-P | 38.41 | 92.77 | 100.00 | 65.95 | 67.75 | **82.72** | 87.19 | 66.03 |
| | FS-OPT | 28.90 | 94.12 | 83.87 | 71.83 | 65.20 | 82.39 | 81.33 | 74.67 |
| | Ours | **24.08** | **95.26** | **62.67** | 81.67 | **64.15** | 82.00 | **78.33** | 73.53 |
| Other methods | MSP | 52.66 | 91.42 | 67.01 | 82.86 | 80.40 | 74.75 | 83.97 | 73.07 |
| | ODIN | 32.84 | 91.94 | 69.20 | 69.53 | 62.03 | 82.57 | 78.71 | 66.47 |
| | MLS | 31.93 | 93.51 | 64.50 | 82.97 | 70.71 | 80.18 | 82.41 | 74.52 |
| | ASH-P | 29.39 | 93.98 | 84.39 | 66.93 | 68.21 | 81.11 | 86.73 | 65.27 |
| | ASH-B | 28.21 | 94.27 | 93.93 | 53.00 | 57.45 | 83.80 | 93.63 | 57.20 |
| | ASH-S | 23.93 | 94.41 | 82.57 | 68.02 | 52.41 | 84.65 | 89.39 | 59.63 |
| | DICE | 29.67 | 93.27 | 96.64 | 52.17 | 59.56 | 82.26 | 95.78 | 44.48 |

*Table 2.* Performance comparison on CIFAR10/100 datasets. Lower FP is better, higher AU is better. Bold values highlight the best results among element-wise feature shaping methods. Underlined values indicate best results among other methods.

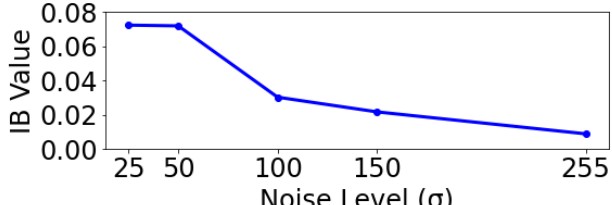

*Figure 5.* IB for the optimal hyperparameter optimized shaping function as a function of the noise level of OOD data. Lower IB corresponds to a more regularized shaping function.

and a regularization based on the Information Bottleneck. We have related the optimal features to several element-wise OOD shaping functions that are used in existing practice, offering a theoretical explanation and suggesting underlying distributional assumptions made in these often empirically motivated approaches. Our theory was shown to lead to a new shaping function that out-performs existing shaping functions on benchmark datasets.

There are several areas for future investigation. Firstly, we have developed the concept of random features, whose mean value models OOD shaping functions, but only exploited the mean value algorithmically. Future work will aim to exploit the OOD feature distribution. Secondly, our theory so far only explains the OOD feature and not the score function. We wish to incorporate scores into our theory. Finally, we wish to explore more general vector shaping functions.

## Impact Statement

This paper presents work whose goal is to advance the field of Machine Learning. There are many potential societal consequences of our work, none which we feel must be specifically highlighted here.

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

## A. Analysis of Loss Function in 1D Gaussian Case

We analytically study the optimization of our loss functional in the case where the ID/OOD distributions of $z$ are Gaussian, i.e., $p(z|y) \sim \mathcal{N}(\mu_y, \sigma_y)$, $y \in \{0, 1\}$, and the feature is a Gaussian with mean being a linear function and the standard deviation constant, i.e., $p(\tilde{z}|z) \sim \mathcal{N}(Wz + b, \sigma_c)$ where $W, b \in \mathbb{R}$. The $W, b$ are parameters that are to be optimized, which specify the OOD feature. This is a relaxation of deterministic shaping function being a linear function. The loss function shape and various components of the loss are shown in Figure 6. The plot shows the loss terms versus $W$; as will be shown below the loss does not depend on $b$. It shows that the separation between feature distributions (KL term) increases as $|W| \to \infty$. On the otherhand, the information bottleneck term increases as $|W| \to 0$. Thus, these terms compete with each other and the optimal solution is well-defined at a finite $|W| > 0$. This simple example suggests that the loss functional is well defined (i.e., has a finite optimum). Notice that without the IB term, there is no optimal value of $W$.

Next, we derive the components of the loss in analytic form. Based on the assumptions specified in the previous paragraph, the formulas for the assumed probabilities are:

$$p(z|y) = \frac{1}{\sqrt{2\pi}\sigma_y} \exp\left(-\frac{1}{2\sigma_y^2}(z - \mu_y)^2\right) \tag{18}$$

$$p(\tilde{z}|z) = \frac{1}{\sqrt{2\pi}\sigma_c} \exp\left(-\frac{(\tilde{z} - Wz - b)^2}{2\sigma_c^2}\right). \tag{19}$$

Under these assumptions, we can calculate the feature distributions:

$$p(\tilde{z}|y) = \frac{1}{\sqrt{2\pi(\sigma_c^2 + W^2\sigma_y^2)}} \exp\left(-\frac{1}{2(\sigma_c^2 + W^2\sigma_y^2)}(\tilde{z} - W\mu_y - b)^2\right), \tag{20}$$

and note that the joint distribution is

$$p(\tilde{z}, z|y) \sim N(\mu_{\tilde{z},z}, \Sigma_{\tilde{z},z}), \quad \mu_{\tilde{z},z} = \begin{pmatrix} W\mu_y + b \\ \mu_y \end{pmatrix}, \quad \Sigma_{z,\tilde{z}} = \begin{pmatrix} \sigma_c^2 + W^2\sigma^2 & W\sigma^2 \\ W\sigma^2 & \sigma^2 \end{pmatrix}. \tag{21}$$

Let us compute the loss of both the KL term and the information bottleneck under this case. First note the formula: if $p \sim \mathcal{N}(\hat{\mu}_1, \hat{\sigma}_1), q \sim \mathcal{N}(\hat{\mu}_2, \hat{\sigma}_2)$ then

$$D_{KL}(p||q) = \log\left(\frac{\hat{\sigma}_2}{\hat{\sigma}_1}\right) + \frac{\hat{\sigma}_1^2 + (\hat{\mu}_1 - \hat{\mu}_2)^2}{2\hat{\sigma}_2^2} - \frac{1}{2}. \tag{22}$$

Choosing

$$\hat{\mu}_1 = W\mu_1 + b, \tag{23}$$

$$\hat{\mu}_2 = W\mu_0 + b, \tag{24}$$

$$\hat{\sigma}_1^2, \hat{\sigma}_2^2 = \sigma_c^2 + W^2\sigma^2, \tag{25}$$

we find that

$$D_{KL}(p(\tilde{z}|Y=1) \,||\, p(\tilde{z}|Y=0)) = \frac{\sigma_c^2 + W^2\sigma^2 + W^2(\mu_1 - \mu_0)^2}{2(\sigma_c^2 + W^2\sigma^2)} - \frac{1}{2} = \frac{W^2(\mu_1 - \mu_0)^2}{2(\sigma_c^2 + W^2\sigma^2)}. \tag{26}$$

Let us now compute the information bottleneck term. Note the following result: if $X, Y \sim \mathcal{N}(\mu, \Sigma)$, where

$$\Sigma = \begin{pmatrix} \sigma_X^2 & \rho\sigma_X\sigma_Y \\ \rho\sigma_X\sigma_Y & \sigma_Y^2 \end{pmatrix}, \tag{27}$$

then

$$I(X; Y) = -\frac{1}{2}\log\left(1 - \rho^2\right). \tag{28}$$

Note that

$$\Sigma_{Z,\tilde{Z}} = \begin{pmatrix} \sigma_c^2 + W^2\sigma^2 & W\sigma^2 \\ W\sigma^2 & \sigma^2 \end{pmatrix}; \tag{29}$$

therefore,

$$\rho_{\tilde{Z},Z} = \frac{W\sigma}{\sqrt{\sigma_c^2 + W^2\sigma^2}} \tag{30}$$

Thus,

$$I(\tilde{Z}; Z) = -\frac{1}{2}\log\left[1 - \frac{W^2\sigma^2}{\sigma_c^2 + W^2\sigma^2}\right] = -\frac{1}{2}\log\left(\frac{\sigma_c^2}{\sigma_c^2 + W^2\sigma^2}\right). \tag{31}$$

Next, we compute

$$I(\tilde{Z}; Y) = p(Y=1)\int p(\tilde{z}|Y=1)\log\frac{p(\tilde{z}|Y=1)}{p(\tilde{z})}\,d\tilde{z} + p(Y=0)\int p(\tilde{z}|Y=0)\log\frac{p(\tilde{z}|Y=0)}{p(\tilde{z})}\,d\tilde{z} \tag{32}$$

$$= -p(Y=1)h(p(\tilde{z}|Y=1)) - p(Y=0)h(p(\tilde{z}|Y=0)) + h(p(\tilde{z})) \tag{33}$$

$$= -\frac{1}{2}\left[\log\left(2\pi(\sigma_c^2 + W^2\sigma^2)\right) + 1\right] + h(p(\tilde{z})), \tag{34}$$

where we used that

$$h(\mathcal{N}(\mu, \sigma^2)) = \frac{1}{2}\left[\log(2\pi\sigma^2) + 1\right]. \tag{35}$$

Note that

$$p(\tilde{z}) = p(Y=1)p(\tilde{z}|Y=1) + p(Y=0)p(\tilde{z}|Y=0) \tag{36}$$

$$= p(Y=1)G(\tilde{z}'; W\mu_0 + b, \sigma_{\tilde{z}}) + p(Y=0)G(\tilde{z}'; W\mu_1 + b, \sigma_{\tilde{z}}) \tag{37}$$

$$= \frac{1}{\sigma_{\tilde{z}}}\left[p(Y=1)G(z'; 0, 1) + p(Y=0)G(z'; \mu', 1)\right], \tag{38}$$

where

$$z' = \frac{\tilde{z} - W\mu_0 - b}{\sigma_{\tilde{z}}}, \quad \sigma_{\tilde{z}}^2 = \sigma_c^2 + W^2\sigma^2, \quad \mu' = \frac{W(\mu_1 - \mu_0)}{\sigma_{\tilde{z}}}. \tag{39}$$

Thus,

$$h(p(\tilde{z})) = -\int p(\tilde{z})\log p(\tilde{z})\,d\tilde{z} \tag{40}$$

$$= -\sigma_{\tilde{z}}\int p(z')\log p(z')\,dz' \tag{41}$$

$$= \int \tilde{G}(z')\log\left[\frac{1}{\sigma_{\tilde{z}}}\tilde{G}(z')\right]dz' \tag{42}$$

$$= \log\sigma_{\tilde{z}} - \int \tilde{G}(z')\log\tilde{G}(z')\,dz' \tag{43}$$

$$= \log\sigma_{\tilde{z}} + h(\tilde{G}), \tag{44}$$

where

$$\tilde{G}(z') = p(Y=1)G(z'; 0, 1) + p(Y=0)G(z'; \mu', 1). \tag{45}$$

Therfore,

$$I(\tilde{Z}; Y) = -\frac{1}{2}\left[\log(2\pi\sigma_{\tilde{z}}^2) + 1\right] + \log\sigma_{\tilde{z}} + h(\tilde{G}) \tag{46}$$

$$= -\frac{1}{2}\left[\log(2\pi) + 1\right] + h(\tilde{G}). \tag{47}$$

Therefore,

$$\text{IB}(p(\tilde{z}|z)) = \log\left(\frac{\sigma_{\tilde{z}}}{\sigma_c}\right) + \frac{\beta}{2}\left[\log(2\pi) + 1\right] - \beta h(\tilde{G}). \tag{48}$$

Finally,

$$L(p(\tilde{z}|z)) = -\frac{W^2(\mu_1 - \mu_0)^2}{2\sigma_{\tilde{z}}^2} + \log\left(\frac{\sigma_{\tilde{z}}}{\sigma_c}\right) + \frac{\beta}{2}\left[\log(2\pi) + 1\right] - \beta h(\tilde{G}), \tag{49}$$

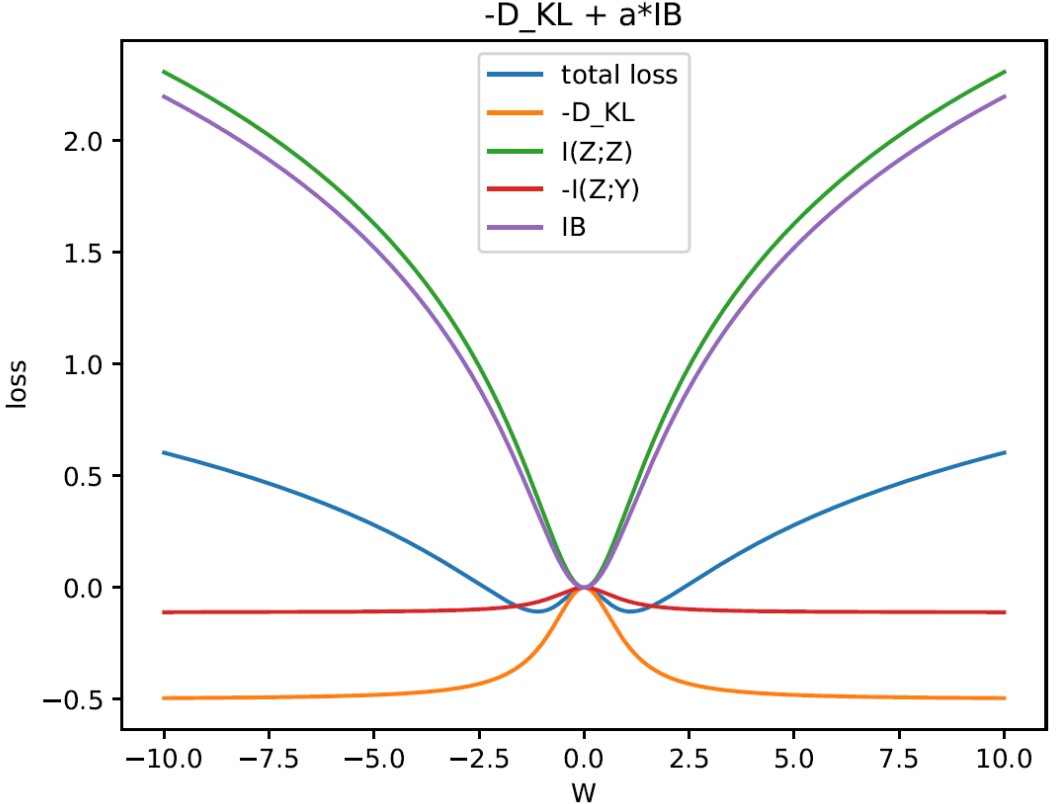

*Figure 6.* Loss function for the 1D Gaussian case. Note $\mu_0 - \mu_1 = 1$, $\sigma = \sigma_c = 1$, and $\beta = 1$. Note $L = D_{KL} + \alpha\text{IB}$ where $\alpha = 0.5$.

where

$$\tilde{G}(z) = p(Y = 1)G(z; 0, 1) + p(Y = 0)G(z; \mu', 1) \tag{50}$$

$$\sigma_{\tilde{z}}^2 = \sigma_c^2 + W^2\sigma^2 \tag{51}$$

$$\mu' = \frac{W(\mu_0 - \mu_1)}{\sigma_{\tilde{z}}}. \tag{52}$$

We show the plot of this loss function in Figure 6.

## B. Simplifying the Loss With Independence Assumptions

We simplify our loss functional under independence assumptions. Specifically, we assume that

$$p(\tilde{z}|z) = \prod_{i=1}^{n} p(\tilde{z}_i|z_i) \tag{53}$$

$$p(z|y) = \prod_{i=1}^{n} p(z_i|Y = y). \tag{54}$$

Note that

$$p(\tilde{z}|y) = \int p(\tilde{z}|z)p(z|y)\,\mathrm{d}z \tag{55}$$

$$= \int \prod_i p(\tilde{z}_i|z_i)p(z_i|y)\,\mathrm{d}z_1 \ldots \mathrm{d}z_n \tag{56}$$

$$= \prod_i \int p(\tilde{z}_i|z_i)p(z_i|y)\,\mathrm{d}z_i \tag{57}$$

$$= \prod_i p(\tilde{z}_i|y). \tag{58}$$

Therefore,

$$D_{KL}[p(\tilde{z}|0)\,||\,p(\tilde{z}|1)] = \int p(\tilde{z}|0)\log\prod_i \frac{p(\tilde{z}_i|0)}{p(\tilde{z}_i|1)}\,\mathrm{d}\tilde{z} \tag{59}$$

$$= \sum_i \int p(\tilde{z}|0)\log\frac{p(\tilde{z}_i|0)}{p(\tilde{z}_i|1)}\,\mathrm{d}\tilde{z} \tag{60}$$

$$= \sum_i D_{KL}[p(\tilde{z}_i|0)\,||\,p(\tilde{z}_i|1)], \tag{61}$$

and by a similar computation, we get that

$$I(\tilde{Z};Z) = \sum_i I(\tilde{Z}_i;Z_i) \tag{62}$$

$$I(\tilde{Z};Y) = \sum_i I(\tilde{Z}_i;Y). \tag{63}$$

Thus, we see that

$$L(p(\tilde{z}|z)) = -D_KL[p(\tilde{z}|0)\,||\,p(\tilde{z}|1)] + \alpha\mathbf{IB}(p(\tilde{z}|z)) = \sum_i L_i(p(\tilde{z}_i|z_i)), \tag{64}$$

where

$$L_i(p(\tilde{z}_i|z_i)) = -D_{KL}[p(\tilde{z}_i|0)\,||\,p(\tilde{z}_i|1)] + \alpha[I(\tilde{Z}_i;Z_i) - \beta I(\tilde{Z}_i;Y)]. \tag{65}$$

Therefore, we just need to solve $n$ independent problems:

$$\underset{p(\tilde{z}_i|z_i)}{\arg\min}\, L_i(p(\tilde{z}_i|z_i)), \quad i \in \{1,\ldots,n\}. \tag{66}$$

## C. Gradient of Loss Computations

We review the definition of gradient of functionals, which are functions defined on functions, in order to compute gradients of our loss. First, we define the directional derivative. Let $\delta p(\tilde{z}|z)$ denote a perturbation of $p(\tilde{z}|z)$, which is a function of $\tilde{z}$ with integral zero so that $p(\tilde{z}|z) + \varepsilon\delta p(\tilde{z}|z)$ defines a valid probability (i.e., integrates to 1) for $\varepsilon$ small. The direction derivative is defined as

$$\mathrm{d}L(p(\tilde{z}|z)) \cdot \delta p(\tilde{z}|z) = \left.\frac{\mathrm{d}}{\mathrm{d}\varepsilon}\, L[p(\tilde{z}|z) + \varepsilon\delta p(\tilde{z}|z)]\right|_{\varepsilon=0}. \tag{67}$$

The gradient $\nabla_{p(\tilde{z}|z)}L(p(\tilde{z}|z))$ is the perturbation of $p(\tilde{z}|z)$ that satisfies the relation:

$$\mathrm{d}L(p(\tilde{z}|z)) \cdot \delta p(\tilde{z}|z) = \int \nabla_{p(\tilde{z}|z)}L(p(\tilde{z}|z)) \cdot \delta p(\tilde{z}|z)\,\mathrm{d}z. \tag{68}$$

## C.1. KL Loss Gradient

We look into the optimization of

$$\max_{p(\tilde{z}|z)} D_{KL}[p(\tilde{z}|Y=1) \, || \, p(\tilde{z}|Y=0)], \tag{69}$$

where $D_{KL}$ is the KL-divergence or relative entropy:

$$D_{KL}[p \, || \, q] = \int p(x) \log \frac{p(x)}{q(x)} \, \mathrm{d}x, \tag{70}$$

that is we would like to compute $\tilde{Z}$ such that the resulting distributions under in/out data are maximally separated.

We compute the optimizing conditions by computing the variation. First note the following:

$$p(\tilde{z}|y) = \int p(\tilde{z}|z, y) p(z|y) \, \mathrm{d}z = \int p(\tilde{z}|z) p(z|y) \, \mathrm{d}z, \tag{71}$$

where the equality on the right hand side is by assumption - we do not want our feature $\tilde{Z}$ to be dependent on whether the data is OOD or not. Now we compute the variation:

$$\delta D_{KL} \cdot \delta p(\tilde{z}|z) = \int \delta p(\tilde{z}|Y=1) \cdot \delta p(\tilde{z}|z) \log \left( \frac{p(\tilde{z}|Y=1)}{p(\tilde{z}|Y=0)} \right) \tag{72}$$

$$+ p(\tilde{z}|Y=0) \delta \left[ \frac{p(\tilde{z}|Y=1)}{p(\tilde{z}|Y=0)} \right] \cdot \delta p(\tilde{z}|z) \, \mathrm{d}\tilde{z}. \tag{73}$$

Let's compute

$$\delta p(\tilde{z}|y) \cdot \delta p(\tilde{z}|z) = \int \delta p(\tilde{z}|z) p(z|y) \, \mathrm{d}z. \tag{74}$$

Therefore,

$$\delta \left[ \frac{p(\tilde{z}|Y=1)}{p(\tilde{z}|Y=0)} \right] \cdot \delta p(\tilde{z}|z) = \frac{\delta p(\tilde{z}|Y=1) p(\tilde{z}|Y=0) - p(\tilde{z}|Y=1) \delta p(\tilde{z}|Y=0)}{p(\tilde{z}|Y=0)} \tag{75}$$

$$= \frac{\int \delta p(\tilde{z}|z) p(z|Y=1) \, \mathrm{d}z \cdot p(\tilde{z}|Y=0) - \int \delta p(\tilde{z}|z) p(z|Y=0) \, \mathrm{d}z \cdot p(\tilde{z}|Y=1)}{p(\tilde{z}|Y=0)^2}. \tag{76}$$

Therefore,

$$p(\tilde{z}|Y=0) \delta \left[ \frac{p(\tilde{z}|Y=1)}{p(\tilde{z}|Y=0)} \right] \cdot \delta p(\tilde{z}|z) = \int \delta p(\tilde{z}|z) \left[ p(z|Y=1) - L(\tilde{z}) p(z|Y=0) \right] \, \mathrm{d}z, \tag{77}$$

where

$$l(\tilde{z}) = \frac{p(\tilde{z}|Y=1)}{p(\tilde{z}|Y=0)}, \tag{78}$$

is the likelihood ratio of the distributions of $\tilde{Z}$. Thus,

$$\delta D_{KL} \cdot \delta p(\tilde{z}|z) = \int \int \delta p(\tilde{z}|z) \left[ (1 + \log l(\tilde{z})) \, p(z|Y=1) - l(\tilde{z}) p(z|Y=0) \right] \, \mathrm{d}z \, \mathrm{d}\tilde{z}, \tag{79}$$

where

$$l(z) = \frac{p(z|Y=1)}{p(z|Y=0)} = \frac{p_{in}(z)}{p_{out}(z)}. \tag{80}$$

Therefore,

$$\nabla_{p(\tilde{z}|z)} D_{KL} = p_{out}(z) \cdot \left[ (1 + \log l(\tilde{z})) l(z) - l(\tilde{z}) \right]. \tag{81}$$

## C.2. Information Bottleneck Gradient

We consider the information bottleneck term:

$$\text{IB}(p(\tilde{z}|z)) = I(Z; \tilde{Z}) - \beta I(\tilde{Z}; Y), \tag{82}$$

where $I$ denotes mutual information. Note that

$$I(Z; \tilde{Z}) = \int p(z, \tilde{z}) \log \frac{p(z, \tilde{z})}{p(z)p(\tilde{z})} \, \mathrm{d}\tilde{z} \, \mathrm{d}z = \int p(z)p(\tilde{z}|z) \log \frac{p(\tilde{z}|z)}{p(\tilde{z})} \, \mathrm{d}\tilde{z} \, \mathrm{d}z. \tag{83}$$

Also,

$$I(\tilde{Z}; Y) = \sum_{y \in \{0,1\}} \int p(\tilde{z}, y) \log \frac{p(\tilde{z}, y)}{p(\tilde{z})p(y)} \, \mathrm{d}\tilde{z} = \sum_{y \in \{0,1\}} \int p(\tilde{z}|y)p(y) \log \frac{p(\tilde{z}|y)}{p(\tilde{z})} \, \mathrm{d}\tilde{z}. \tag{84}$$

Let us compute the variation of these terms:

$$\delta I(Z; \tilde{Z}) \cdot \delta p(\tilde{z}|z) = \int p(z)\delta p(\tilde{z}|z) \log \frac{p(\tilde{z}|z)}{p(\tilde{z})} + p(z)\frac{\delta p(\tilde{z}|z)p(\tilde{z}) - p(\tilde{z}|z)\delta p(\tilde{z}) \cdot \delta p(\tilde{z}|z)}{p(\tilde{z})} \, \mathrm{d}\tilde{z} \, \mathrm{d}z \tag{85}$$

$$= \int \delta p(\tilde{z}|z)p(z) \left[ 1 + \log \frac{p(\tilde{z}|z)}{p(\tilde{z})} \right] - \frac{p(\tilde{z}|z)p(z)}{p(\tilde{z})} \int \delta p(\tilde{z}|z')p(z') \, \mathrm{d}z' \, \mathrm{d}\tilde{z} \, \mathrm{d}z. \tag{86}$$

Let us evaluate the term after the minus sign:

$$\int \int \int \delta p(\tilde{z}|z')\frac{p(\tilde{z}|z)p(z)}{p(\tilde{z})}p(z') \, \mathrm{d}z' \, \mathrm{d}\tilde{z} \, \mathrm{d}z = \int \int \delta p(\tilde{z}|z')\frac{p(z')}{p(\tilde{z})} \int p(\tilde{z}|z)p(z) \, \mathrm{d}z \, \mathrm{d}\tilde{z} \, \mathrm{d}z' \tag{87}$$

$$= \int \int \delta p(\tilde{z}|z')p(z') \, \mathrm{d}\tilde{z} \, \mathrm{d}z'. \tag{88}$$

Therefore,

$$\delta I(Z; \tilde{Z}) \cdot \delta p(\tilde{z}|z) = \int \delta p(\tilde{z}|z)p(z) \log \frac{p(\tilde{z}|z)}{p(\tilde{z})} \, \mathrm{d}\tilde{z} \, \mathrm{d}z, \tag{89}$$

$$\nabla_{p(\tilde{z}|z)} I(Z; \tilde{Z}) = p(z) \log \frac{p(\tilde{z}|z)}{p(\tilde{z})}. \tag{90}$$

Let us compute the variation of the second term in IB:

$$\delta I(\tilde{Z}; Y) = \sum_{y \in \{0,1\}} \int \delta p(\tilde{z}|y)p(y) \log \frac{p(\tilde{z}|y)}{p(\tilde{z})} + p(y)\frac{\delta p(\tilde{z}|y)p(\tilde{z}) - p(\tilde{z}|y)\delta p(\tilde{z})}{p(\tilde{z})} \, \mathrm{d}z. \tag{91}$$

Note that

$$\delta p(\tilde{z}) = \int \delta p(\tilde{z}|z)p(z) \, \mathrm{d}z \tag{92}$$

$$\delta p(\tilde{z}|y) = \int \delta p(\tilde{z}|z)p(z|y) \, \mathrm{d}\tilde{z}. \tag{93}$$

Therefore,

$$\delta I(\tilde{Z}; Y) = \sum_{y \in \{0,1\}} \int \int \delta p(\tilde{z}|z)p(y) \left[ p(z|y) \log \frac{p(\tilde{z}|y)}{p(\tilde{z})} + p(z|y) - \frac{p(\tilde{z}|y)p(z)}{p(\tilde{z})} \right] \, \mathrm{d}\tilde{z} \, \mathrm{d}z \tag{94}$$

$$= \int \int \delta p(\tilde{z}|z) \sum_{y \in \{0,1\}} p(y) \left[ p(z|y) \log \frac{p(\tilde{z}|y)}{p(\tilde{z})} + p(z|y) - \frac{p(\tilde{z}|y)p(z)}{p(\tilde{z})} \right] \, \mathrm{d}\tilde{z} \, \mathrm{d}z \tag{95}$$

$$= \int \int \delta p(\tilde{z}|z) \sum_{y \in \{0,1\}} p(y)p(z|y) \log \frac{p(\tilde{z}|y)}{p(\tilde{z})} \, \mathrm{d}\tilde{z} \, \mathrm{d}z. \tag{96}$$

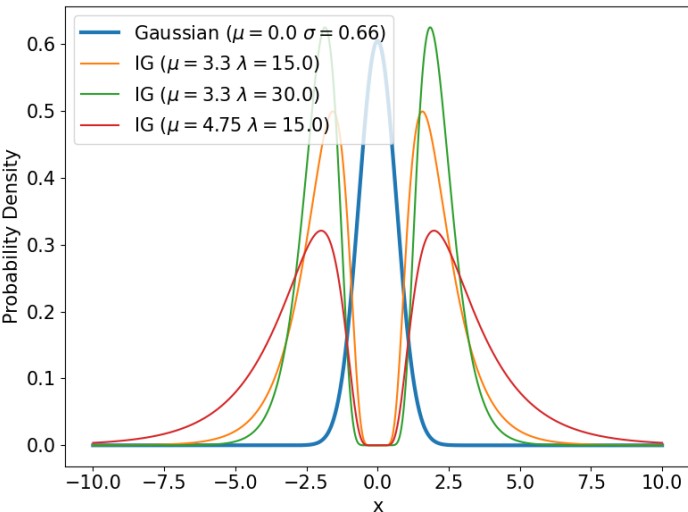

*Figure 7.* Plot of Inverse Gaussian (IG) distribution, $p(z|1) \sim IG(d(z); \mu, \lambda)$, under different parameters with a Gaussian (blue). Note that IG has high probability where the Gaussian does not.

Therefore,

$$\nabla_{p(\tilde{z}|z)} \mathrm{IB} = p(z) \log \frac{p(\tilde{z}|z)}{p(\tilde{z})} - \beta \sum_{y \in \{0,1\}} p(y) p(z|y) \log \frac{p(\tilde{z}|y)}{p(\tilde{z})} \tag{97}$$

$$= \sum_{y \in \{0,1\}} p(y) p(z|y) \left[ \log \frac{p(\tilde{z}|z)}{p(\tilde{z})} - \beta \log \frac{p(\tilde{z}|y)}{p(\tilde{z})} \right]. \tag{98}$$

## D. Plot of Inverse Gaussian Distribution

In Figure 7, we show a plot of our Inverse Gaussian for various parameters along with a Gaussian. Notice that the IG has mass complementary to the Gaussian, and thus represents a natural distribution for OOD if the ID is Gaussian.

## E. A Study of Our Feature Shaping Over Parameters

In this section, we perform an extended study of feature shaping as a function of parameters of the distributions and the $\beta$ parameter in the Information Bottleneck, extending the study in Section 4 of the main paper.

### E.1. Feature Shaping Versus $\beta$

In Figure 8, we explore how our feature shaping function varies as a function of $\beta$ for various distributions.

### E.2. Feature Shaping Versus Distribution Parameters

In Figure 9, we explore how our feature shaping function varies with different distribution parameters.

## F. Visualizations for Additive Gaussian Noise Experiment

Figure 10 shows a visualization of an example ImageNet image and various noise levels added to it to simulate OOD data for the experiment in Section 6.

Figure 11 visualizes the activations of images from the validation set of ImageNet-1k with different levels of additive Gaussian noises. From figure 11a, with more noises, the activations are more instable and have more larger spikes. From figure 11b, the activations shrink more and peak more at small values.

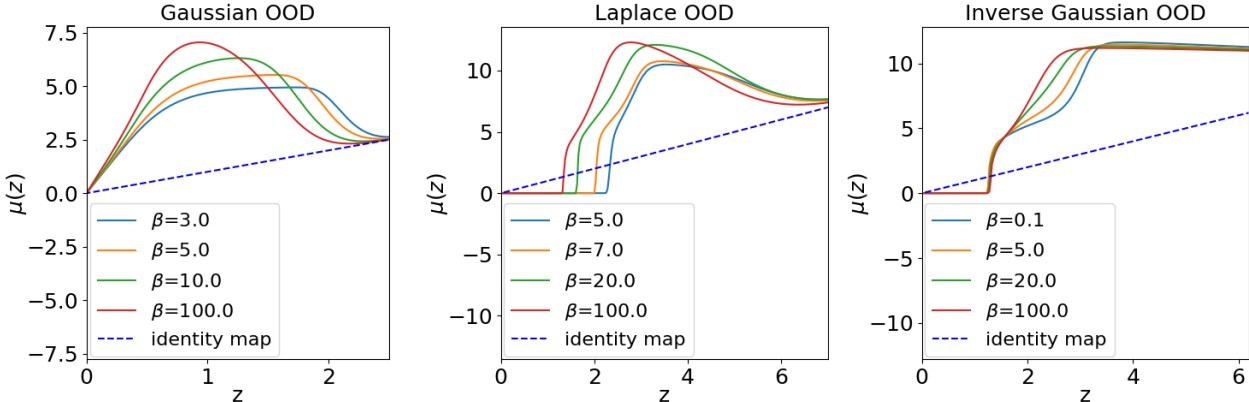

*Figure 8.* The mean of the OOD Gaussian Random Feature under the Gaussian (left), Laplace (middle) and Inverse Gaussian (right) distributions for the OOD distribution. Different curves on the same plot indicate differing weights on the $I(\tilde{Z}; Y)$ component of the Information Bottleneck term, $\beta$. The weight on the IB is fixed to $\alpha = 3.0$. For the Gaussian case, $p(z|0) \sim \mathcal{N}(-0.5, 0.5)$ and $p(z|1) \sim \mathcal{N}(0.5, 0.5)$. For the Laplace case, $p(z|0) \sim \mathcal{N}(0, 0.5)$ and $p(z|1) \sim Lap(0, 1)$. In the Inverse Gaussian case, $p(z|0) \sim \mathcal{N}(0, 0.5)$ and $p(z|1) \sim IG(d(z); 0.5, 15)$.

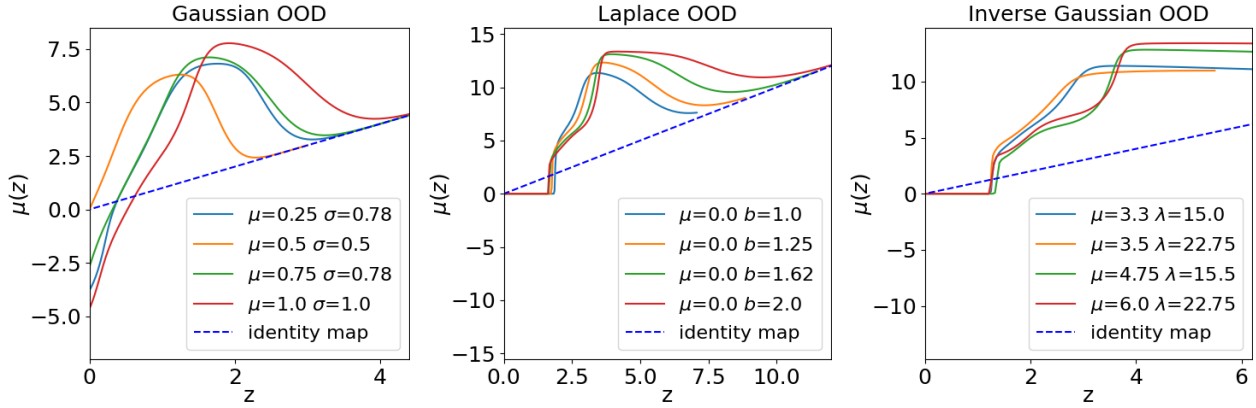

*Figure 9.* The mean of the OOD Gaussian Random Feature under the Gaussian (left), Laplace (middle) and Inverse Gaussian (right) distributions for the OOD distribution. Different curves on the same plot indicate different OOD distribution parameters. The weight on the IB term and weight of its $I(\tilde{Z}; Y)$ component are set as $\alpha = 3.0$ and $\beta = 10.0$. For the Gaussian case, $p(z|0) \sim \mathcal{N}(-0.5, 0.5)$. For the Laplace case, $p(z|0) \sim \mathcal{N}(0, 0.5)$. In the Inverse Gaussian case, $p(z|0) \sim \mathcal{N}(0, 0.5)$.

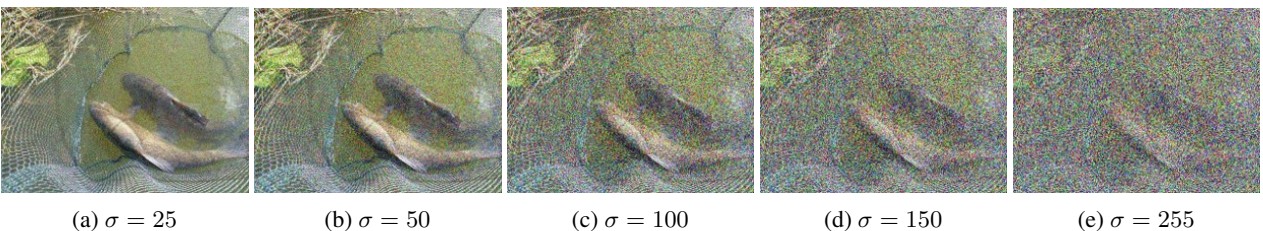

(a) $\sigma = 25$      (b) $\sigma = 50$      (c) $\sigma = 100$      (d) $\sigma = 150$      (e) $\sigma = 255$

*Figure 10.* Visualization of a sample image from ImageNet validation split under different levels of noise corruption ($\sigma$ values).

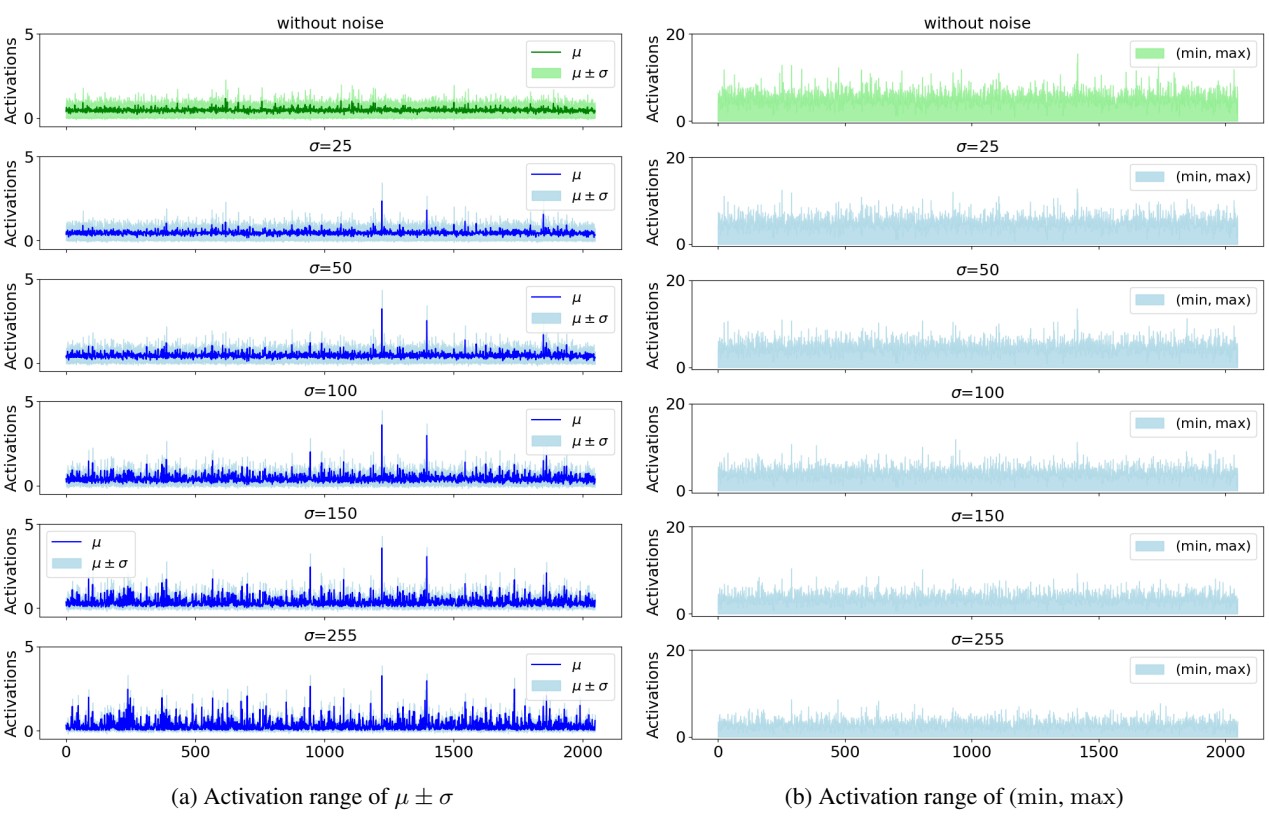

(a) Activation range of $\mu \pm \sigma$

(b) Activation range of $(\min, \max)$

*Figure 11.* The network activations of ImageNet with different levels of additive Gaussian noises. The shaded regions in (a) represent one standard deviation above and below the mean, while those in (b) represent the range of min and max values of activations. With more noises, activations tend to more instable, smaller range of activation values.

## G. Optimal hyperparameters used in ImageNet and CIFAR benchmarking

We report the optimal hyperparameters for the experiments studied in Section 6 in Table 3.

| Model | ID Data | Hyperparameters | | | | | | |
|---|---|---|---|---|---|---|---|---|
| | | $y_0$ | $y_{1a}$ | $z_1$ | $y_{1b}$ | $m_1$ | $z_2$ | $m_2$ |
| ResNet-50 | ImageNet-1k | 0.0 | 0.0 | 0.52 | 0.73 | 0.61 | 1.2 | -0.3 |
| MobileNet-v2 | ImageNet-1k | 0.0 | 0.0 | 0.55 | 0.5 | 0.79 | 1.49 | -0.74 |
| ViT-B-16 | ImageNet-1k | 0.0 | 0.0 | 0.05 | 1.58 | 2.0 | 2.0 | -1.0 |
| ViT-L-16 | ImageNet-1k | 0.0 | 0.0 | 0.06 | 1.76 | 1.79 | 2.0 | -0.32 |
| DenseNet101 | CIFAR 10 | 0.0 | 0.0 | 0.51 | 0.41 | 1.18 | 1.1 | 0.37 |
| MLP-N | CIFAR 10 | -0.3 | 0.25 | 0.73 | 0.40 | 0.10 | 3.54 | 1.76 |
| DenseNet101 | CIFAR 100 | 0.0 | 0.1 | 1.0 | 2.0 | 0.17 | 1.8 | -0.18 |
| MLP-N | CIFAR 100 | 0.0 | 0.3 | 0.59 | 0.4 | 0.1 | 4.0 | 2.0 |

*Table 3.* Our optimal hyperparameters for different models and datasets.

## H. Empirical Distribution of Network Feature Outputs

In this section, we show the empirical distributions of ID and OOD data for various ImageNet-1k benchmarks. As is shown in the subsequent figures, some benchmarks/architectures resemble the distribution assumptions analyzed in this paper (e.g., Gaussian ID/Gaussian OOD - Figure 12 and Gaussian ID/Laplacian OOD - Figure 13 ), thus showing that these may be realistic assumptions. On the otherhand, some datasets (Figure 14 and Figure 15) exhibit distributions that do not fit the distributional assumptions analyzed in this paper. Nevertheless, our novel shaping function still performs well on these benchmarks, showing that our shaping function may well work even when the data differs from the assumed distributions, which is important in practice as exact distributions may not be known.

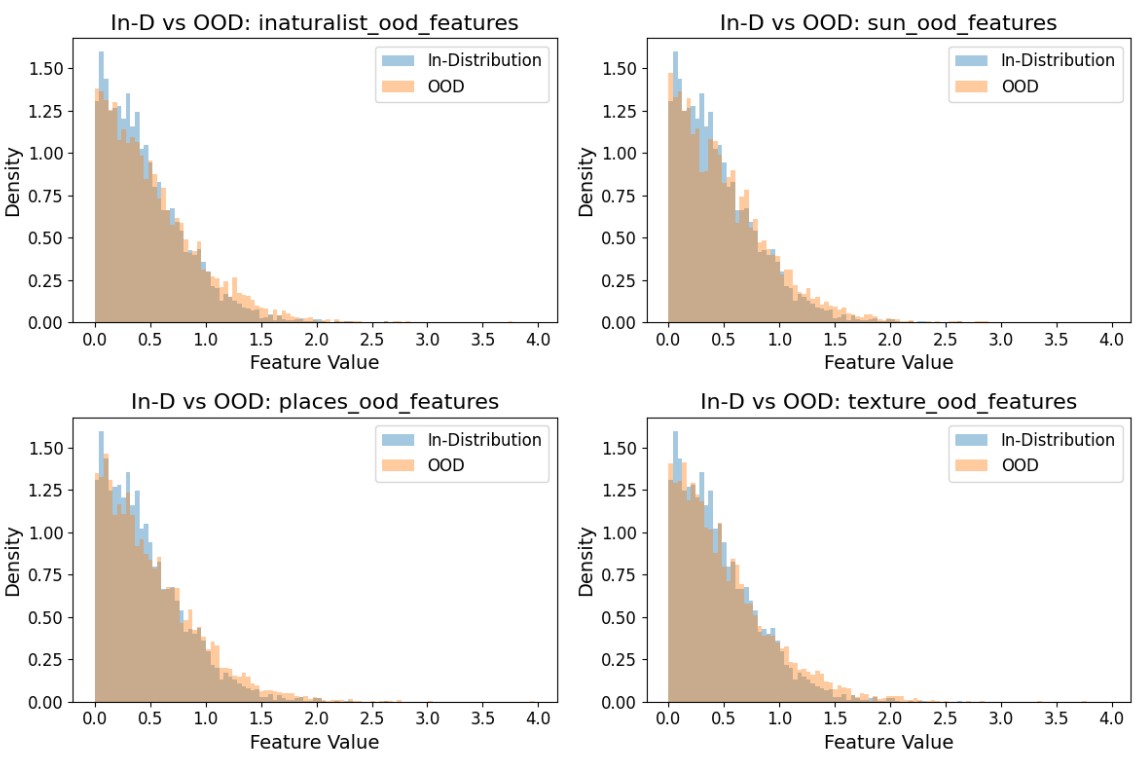

*Figure 12.* Distribution of features from the penultimate layer of ViT-B-16. Comparison with In-distribution data (ImageNet-1k) and different test OOD datasets in the ImageNet-1k benchmark (Zhao et al., 2024). The ID and OOD distributions resemble the positive part of a Gaussian with different variance.

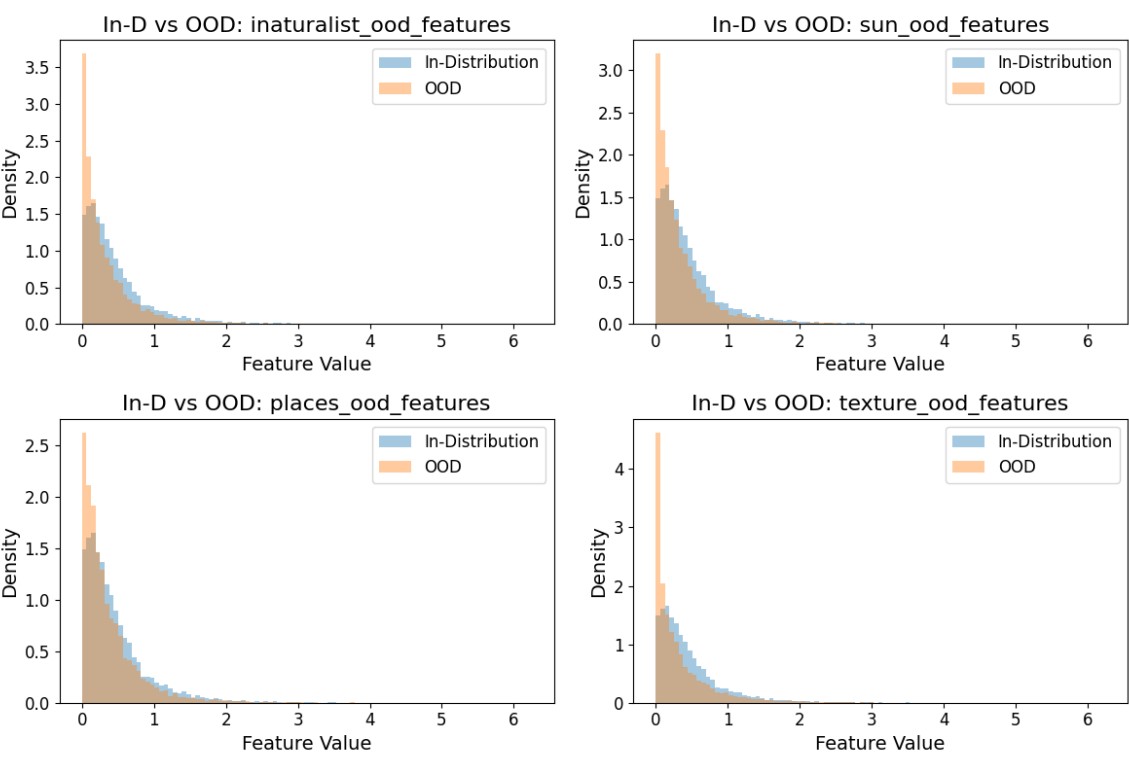

*Figure 13.* Distribution of features from the penultimate layer of ResNet-50. Comparison with In-distribution data (ImageNet-1k) and different test OOD datasets in the ImageNet-1k benchmark (Zhao et al., 2024). The ID distribution resembles the positive part of a Gaussian and the OOD resembles the positive part of a Laplacian distribution.

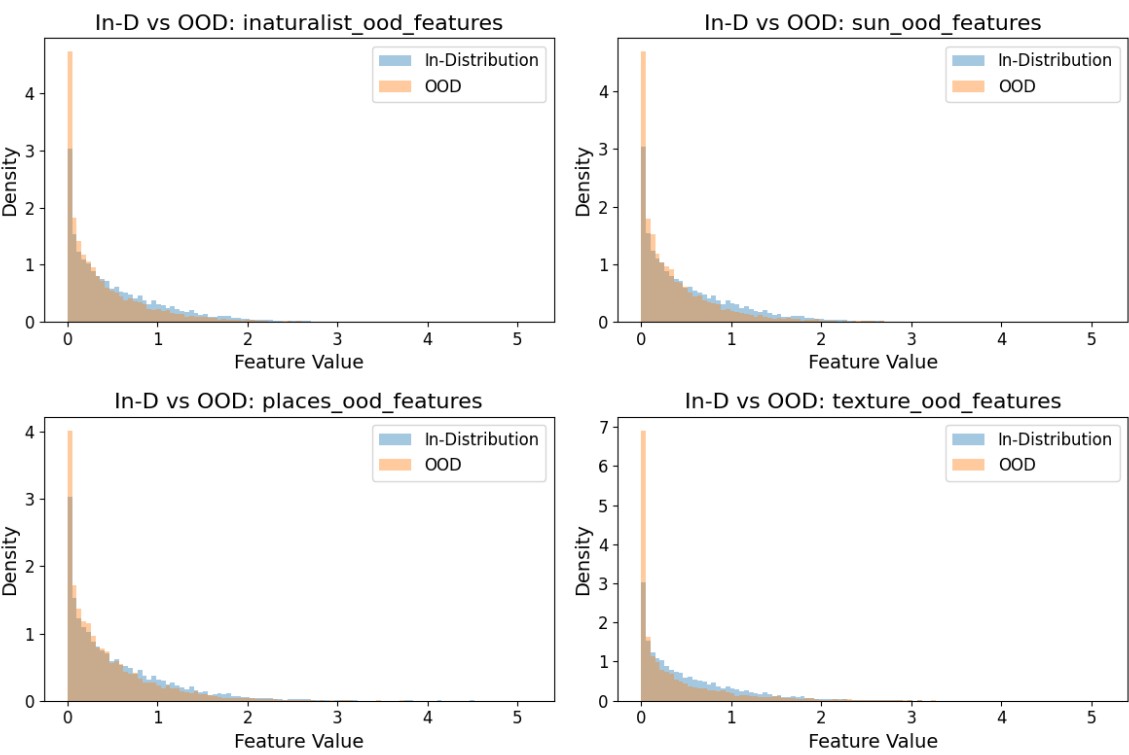

*Figure 14.* Distribution of features from the penultimate layer of MobileNet-V2. Comparison with In-distribution data (ImageNet-1k) and different test OOD datasets in the ImageNet-1k benchmark (Zhao et al., 2024). The ID and OOD both appear Laplacian; although this does not fit the ID assumption analyzed in this paper, our method nevertheless works well on this benchmark.

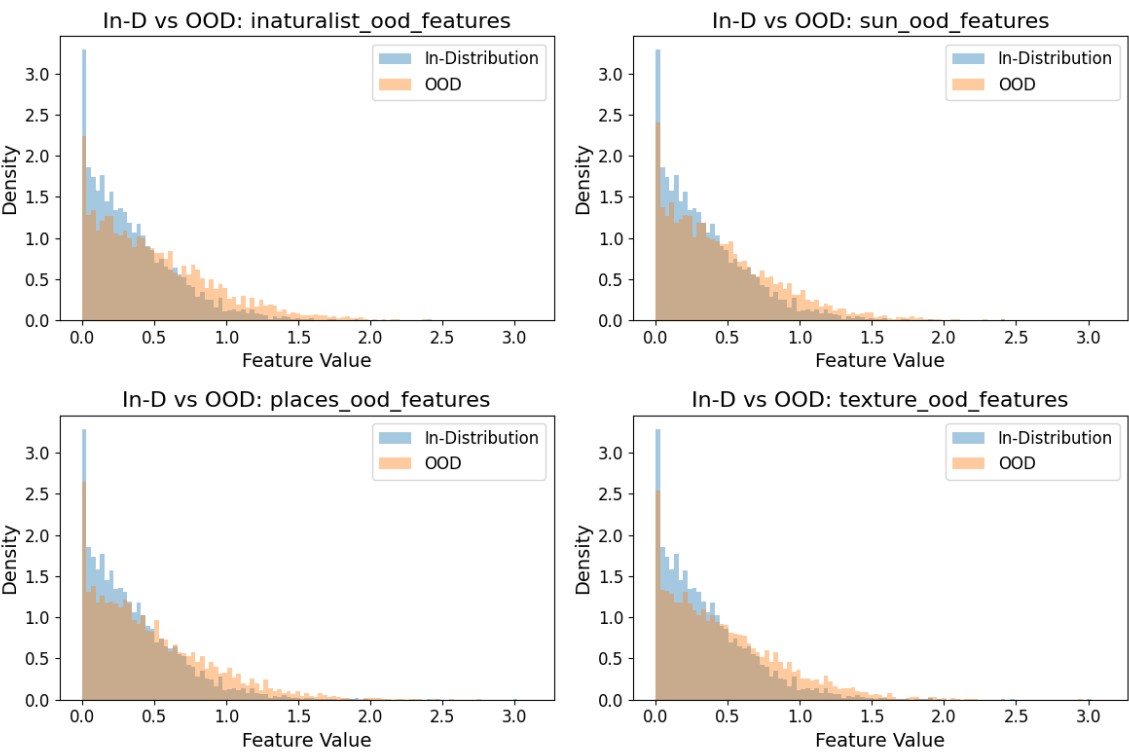

*Figure 15.* Distribution of features from the penultimate layer of ViT-L-16. Comparison with In-distribution data (ImageNet-1k) and different test OOD datasets in the ImageNet-1k benchmark (Zhao et al., 2024). The ID and OOD distributions appear Gaussian but with heavy weight on zeros; although these distributions don't fit the distributional assumptions in the cases analyzed in the paper, our method nevertheless performs well on this benchmark.

