# OpenReview forum: "A Variational Information Theoretic Approach to Out-of-Distribution Detection"
_ICML.cc/2025/Conference — ICML 2025 poster_

### Official Review · Reviewer_q3hx · 2025-03-02

**Overall Recommendation:** 3

**Summary:**

The paper introduces a variational and information bottleneck-informed framework for performing feature shaping for  out-of-distribution detection. Namely, the proposed objective maximizes the KL distribution between the distribution of ID features $Z$ and OOD features $\tilde{Z}$ subject to an information bottleneck regularization which seeks to maximize the mutual information between $Z$ and the ID / OOD label $Y$ while minimizing the mutual information between $Z$ and $Z$. The work show qualitatively that the feature-shaping approaches proposed by past work are similar to their objective under the assumption of independent features and different choices for the distribution of the OOD features (e.g., Gaussian with a different mean from ID, Inverse Gaussian, etc.) and regularization coefficient on the information bottleneck term. Then, the work considers a general piecewise linear shaping function for feature shaping and shows that fitting this function to validation OOD data outperforms previously proposed feature shaping OOD detection methods on test OOD data.

**Claims And Evidence:**

1. It's not clear to me how the proposed theory predicts the generalized piecewise linear feature shaping function proposed later in the work, as mentioned in the abstract.
2. Where does the claim "Note this forms a Markov Chain $Y\rightarrowX\rightarrowZ\rightarrow\tilde{Z}" (line 137-138) come from? It seems more like an assumption the authors are making (a reasonable one to me), but might be better written that way.
3. Other claims seem reasonable.

**Essential References Not Discussed:**

To my knowledge, the essential references are discussed.

**Experimental Designs Or Analyses:**

Experimental design seems sound. The qualitative analysis comparing the mean of $\tilde{z}$ under the proposed objective with different assumptions vs. existing feature shaping methods is quite interesting, but ultimately seems to be an approximate relationship based on the shape of the resulting curves; for instance, the actual functions are not the same if you look at the x and y axes' values.

**Methods And Evaluation Criteria:**

The ImageNet-1k and CIFAR-10/100 OOD detection benchmarks make sense for comparing OOD detection methods. The authors also test 4 different pretrained models encompassing convolution- and transformer-based architectures.

**Other Comments Or Suggestions:**

One way to strengthen this connection mentioned in weaknesses above would be to choose assumption values that would result in functions that are close in distance, not just in shape.

**Other Strengths And Weaknesses:**

Strengths:
1. The paper provides a unifying and principled framework to view many disparate recent methods in OOD detection.
2. The analysis from the framework leads to some observations about the properties that seem to work well for feature shaping.

Weaknesses:
1. The link between the settings discussed in the framework and existing methods is qualitative, as the curves match only in approximate shape.

**Questions For Authors:**

My primary questions are:
1. Can the authors explain what they say that the proposed theory predicts a new shaping function that can outperform existing ones? And explain the evidence for the claim?
2.  Could the authors update Figure 2 to show curves that would closely match those in Figure 3 if overlaid using the same axes?

**Relation To Broader Scientific Literature:**

The paper proposes a unifying framework to think about feature shaping methods for OOD detection, enabling more critical examination of existing and new methods in this space.

**Theoretical Claims:**

I skimmed but did not carefully check the algebra for the Gaussian example and gradient derivations.

---

> ### Author Rebuttal · Authors · 2025-03-31
>
> *Question 1: It's not clear to me how the proposed theory predicts the generalized piecewise linear feature shaping function proposed later in the work, as mentioned in the abstract.*
>
> **Response 1:** See Response 2 to Reviewer p7gJ.
>
> *Question 2: Where does the claim "Note this forms a Markov Chain $Y\rightarrowX\rightarrowZ\rightarrow\tilde{Z}" (line 137-138) come from? It seems more like an assumption the authors are making (a reasonable one to me), but might be better written that way.*
>
> **Response 2:** The reviewer is correct, it is an assumption (not a claim) that Y (ID or OOD class) produces data/image X from which the feature Z is computed, and then from the feature Z, the OOD feature Zt is computed.  The reason for spelling this out is to use the Information Bottleneck, which requires this setup.  We will add a comment in the manuscript.
>
> *Question 3: Experimental design seems sound. The qualitative analysis comparing the mean of
>  under the proposed objective with different assumptions vs. existing feature shaping methods is quite interesting, but ultimately seems to be an approximate relationship based on the shape of the resulting curves; for instance, the actual functions are not the same if you look at the x and y axes' values.*
>
> **Response 3:** Yes, the reviewer is correct – the relation between our shaping functions and SOA is qualitative in shape and approximate as discussed in Section 4 (e.g. Line 294 – that under Gaussian OOD the approach is similar to clipping in ReAct; Line 324 – that under Laplacian OOD a positively sloped region for intermediate values is similar to VRA, etc).  It isn’t our goal to precisely numerically match SOA methods – as many are heuristically derived and there is no reason to believe that the optimal solution based on principles will precisely match empirically driven heuristics.  We aim to understand whether SOA practice, though heuristic-driven has similarities of what an optimal principled approach would predict (hence justifying and understanding these heuristics by showing that the heuristics have traits of a principled approach), but this will not be a precise match in numerical values.  Our work is interested in qualitative understanding of SOA such as, why/when should one clip values of the feature (like ReAct)?, why/when is it a good idea to suppress small values (like ASH)?   We believe the answers to these questions give intuition and insights that will help researchers design new methods.
>
> Having said that, the primary reason that our shaping functions are not on the same scale as current SOA shaping functions is because current SOA’s scale is chosen to fit the energy score function (this is mentioned in the footnote of page 6), while our theory does not involve the score since our goal is to understand the feature making minimal assumptions.  It is nevertheless interesting to see that our shapes (in Fig 2), which resemble SOA, do not depend on choice of score, even though SOA has made the energy score assumption and tuned according to that score.  This indicates the shapes to be a universal property regardless of score.  Note to match scales in practice with the energy score as we do in experiments, our piece-wise linear function family encompasses functions that also have similar scale values to SOA methods.
>
>
> *Question 4: Can the authors explain what they say that the proposed theory predicts a new shaping function that can outperform existing ones? And explain the evidence for the claim?*
>
> **Response 4:** The piece-wise linear family of shaping functions introduced are piece-wise linear approximations of the optimal shaping functions in Fig 2 (see Response 2 to Reviewer p7gJ).  The results of the optimal piecewise linear function have out-performed SOA as shown in the Experiments Section 5.
>
> *Question 5: Could the authors update Figure 2 to show curves that would closely match those in Figure 3 if overlaid using the same axes?*
>
> **Response 5:** Please see Response 3.

---

> > ### Comment · Reviewer_q3hx · 2025-04-02
> >
> > Thanks for the response. My question about motivating the specific piecewise linear function still stands, especially in light of reviewer p7gJ pointing out prior work which also fits a more expressive piecewise linear function.

---

> > > ### Author Response · Authors · 2025-04-02
> > >
> > > Please see Response 4 to Reviewer p7gJ.
> > >
> > > We would like to make two points:
> > >
> > > 1)	As noted in the response to Reviewer p7gJ, compared to FS-Opt ([1] cited by Reviewer p7gJ), we are optimizing the loss over a more general class of functions (infinite dimensional functions) rather than restricting the optimization to piece-wise linear functions.  In that sense our framework is actually considering a much larger class of shaping functions than [1].  The functions that optimize the losses both in [1] and our framework narrow down the class of functions to a particular form (because all of them are clearly not useful).  In our case, the general set of functions is narrowed down to a much smaller a set of functions that are optimizers of our loss. These optimized functions (see Fig 2) can be approximated well with the particular piecewise family in Fig 4 (i.e., this piecewise linear family encompasses resulting optimal mean features under Gaussian, Laplace and IG with various hyperparameters alpha, beta and the distribution parameters).  In the case of [1], the general class of piecewise linear functions is narrowed to the one with the expression given in Eqn 14 of their paper.
> > > 2)	With no other constraints, a more expressive piecewise linear function family is not necessarily better in OOD detection performance as the more expressive piecewise linear family can be associated with distributions that may even be outside the class of realistic OOD distributions (note that our piecewise linear function family comes from distributions that have properties of empirical distributions observed from realistic OOD datasets – see Response 2 to Reviewer 4N3p).  Therefore, with no other constraints, a more expressive family may not lead to better OOD detection performance.

---

### Official Review · Reviewer_p7gJ · 2025-03-12

**Overall Recommendation:** 3

**Summary:**

### Background

- This paper works on the OOD detection task.
- Previous methods use different feature shaping functions to reshape features from the penultimate layer of a pre-trained network. They achieve SoTA performance on OOD detection benchmarks, but lack in theoretical evidence and may not generalize to unseen datasets.

### Method

- The authors develop a theory to formulate OOD features, and they propose a new loss functional consisting of two terms: “the first based on the KL divergence separates resulting ID and OOD feature distributions and the second term is the Information Bottleneck, which favors compressed features that retain the OOD information.”
- Their theory can recover properties of existing feature shaping functions, based on different assumptions on OOD distributions.
- They also design a new shaping function.

### Results

- Their new shaping function outperforms previous methods on OOD detection benchmarks.

**Claims And Evidence:**

Yes

**Essential References Not Discussed:**

I think all related works are included and discussed.

**Experimental Designs Or Analyses:**

The experiments are conducted on standard OOD detection benchmarks. The authors validate their method with different models and different datasets.

**Methods And Evaluation Criteria:**

- The proposed method is a piecewise linear shaping function, which contains 7 hyperparameters. According to the supplementary materials, the values of these hyperparameters are varying with different models on different benchmarks. For example, $y_{1b}$ is 0.73 when using ResNet-50 but 1.76 when ViT-L-16. How do the authors choose the values of hyperparameters?
- I think the proposed method is not related with the theoretical analysis. I appologize if I miss something important. I expected that, the authors can "infer" some good shaping functions based on their theory. For example, maybe the theory can tell us what family of functions is good or help us to decide the values of hyperparemeters. But I don't see how the authors utilize their theory to design the function. It seems that the proposed function is mannually designed on different datasets.

**Other Comments Or Suggestions:**

Algorithm 1 looks very sparse. Maybe you can modify it somehow to make it more compact.

**Other Strengths And Weaknesses:**

### Strengths
- Provide theoretical analysis, which is very interesting. The theory can recover properties of previous methods.

- FS-OPT only considered the mean of ID and OOD features, while this work considers the distribution.

- Their proposed method achieves better performance on different benchmarks.

### Weaknesses

- Another major drawback of FS-OPT is the assumption of independence between features, so their framework cannot explain vector-based shaping functions like ASH. This paper also assumes independence of features, so I think it can be extended in the future.

**Questions For Authors:**

I think the theoretical analysis is very interesting but the proposed method is confusing. I will raise my score if the authors can address my concerns.

**Relation To Broader Scientific Literature:**

Feature shaping methods achieve the SoTA performance in OOD detection, but lacks in theoretical analysis. This paper provides a new theory, which can recover some properties of previous shaping functions. I believe their paper can inspire more ideas in the future.

**Theoretical Claims:**

Sorry that there are some parts I don’t quite understand. Could the authors provide more explanation?

- How to understand OOD feature is $\tilde{Z}\sim p(\tilde{z}|z)$? Why does the OOD feature depend on feature $z$? What does this $p$ mean, the neural network or the shaping function?

- Eq (4), how to understand "where \tilde{Z} is analogous to T and Z is analogous to X"?

- I'm not sure if the proposed theory aligns with the real practice. Under different OOD distribution assumptions, the theory can recover some properties of previous methods, but normally, these methods are all good on ImageNet benchmarks. So, what is the real OOD distribution of the benchmark? Which method is the optimal one?

---

> ### Author Rebuttal · Authors · 2025-03-31
>
> *Question 1: The proposed method is a piecewise linear shaping function, which contains 7 hyperparameters. According to the supplementary materials, the values of these hyperparameters are varying with different models on different benchmarks. For example,
> is 0.73 when using ResNet-50 but 1.76 when ViT-L-16. How do the authors choose the values of hyperparameters?*
>
> **Response 1:** See Experiments Section 5 Line 342-350 (2nd column).  Yes, the optimal shaping function will depend on the specific architecture and its weights (hence the ID datasets) since each will have different statistical distributions – these can all be determined in training as only the ID dataset is needed.  Note the hyper-parameters are fixed across all OOD datasets.
>
> *Question 2: I think the proposed method is not related with the theoretical analysis. I appologize if I miss something important. I expected that, the authors can "infer" some good shaping functions based on their theory. For example, maybe the theory can tell us what family of functions is good or help us to decide the values of hyperparemeters. But I don't see how the authors utilize their theory to design the function. It seems that the proposed function is mannually designed on different datasets.*
>
> **Response 2:**  The reviewer’s expectation is correct our theory should predict a good shaping function, and it does.  The piecewise linear family used in the experiments encompasses piecewise linear approximations of the resulting optimal shaping functions from the Gaussian, Laplacian and Inverse Gaussian OOD distributions, as discussed in Section 4 Lines 344-350.  Notice that shaping functions for the Gaussian, Laplace, IG  (Figure 2) all can be approximated well with a function in the family in Fig.4.  So rather than tuning hyper-parameters for the given distribution and searching over the three distributions, we instead equivalently tune the hyper-parameters for the piecewise linear family – as this is more convenient practically.  Note this not unlike previous literature (e.g., VRA) where hyper-parameter tuning of the shaping function is required.
>
> The proposed shaping function family is not manually designed on different datasets, see Response 1.
>
> *Question 3: How to understand OOD feature is
> ? Why does the OOD feature depend on feature
> ? What does this
>  mean, the neural network or the shaping function?*
>
> **Response 3:** $p(\tilde z|z)$ is the conditional probability of the OOD feature ($\tilde z$) given the input standard NN feature (z) – see Lines 125-132.  $p$ is a probability.  Note that just as conventional deterministic shaping functions (e.g., f(z)) depend on the input feature z, so does our random feature; i.e., one needs to input the NN feature to determine the corresponding OOD feature.
>
> *Question 4: Eq (4), how to understand "where \tilde{Z} is analogous to T and Z is analogous to X"?*
>
> **Response 4:** In traditional IB (from Tishby et al 2000) – $T$ is a compressed representation of the data/feature $X$ that we want to solve for, and $Y$ is the random variable whose information is to be preserved from $X$.  The traditional IB objective function is then $IB = I(X;T) – \beta  I(T;Y)$.
> In our formulation, $\tilde Z$ is a compressed representation of the feature $Z$, and $Y$ is the information that we would like to preserve from $Z$. Since $\tilde Z$ is the compressed variable, it is analogous to $T$; and since $Z$ is the original feature/data, it is analogous to $X$. Our IB term is $IB = I(Z;\tilde Z) – \beta  I(\tilde Z;Y)$.
>
>
> *Question 5: I'm not sure if the proposed theory aligns with the real practice. Under different OOD distribution assumptions, the theory can recover some properties of previous methods, but normally, these methods are all good on ImageNet benchmarks. So, what is the real OOD distribution of the benchmark? Which method is the optimal one?*
>
> **Response 5:**  See Reviewer 4N3p Response 2.  There is no “real” distribution of the benchmark as there are many datasets in the benchmark with different distributions.
>
> *Question 6: Algorithm 1 looks very sparse. Maybe you can modify it somehow to make it more compact.*
>
> **Response 6:** Thanks, will do.

---

> > ### Comment · Reviewer_p7gJ · 2025-04-01
> >
> > Thank you! Most of my concerns are solved, and thus, I raised my score to 3.
> >
> > But I'm still confused with the relationship between the theoretical analysis and the proposed method. I re-read the discussion in Section 4 Lines 344-350. It said: *"The above distributions all approximately fit in a piecewise linear function family, so in the experimental section we explore this as shaping functions."*
> >
> > In my understanding, the authors conduct experiments on the Gaussian, Laplacian and Inverse Gaussian OOD distributions, and they observe that all the distributions approximately fit a piecewise linear function family. And thus, they explore this as shaping functions.
> >
> > But I have two main concerns.
> >
> > First, a piecewise linear function can fit any functions, as long as there are enough number of “pieces”. [1] already uses a 100-piece function and optimizes it for OOD detection. What is the advantage of the proposed piecewise linear function over theirs?
> >
> > Second, more importantly, OOD distributions are typically unknown in practice, and they may not fit in any of "the Gaussian, Laplacian and Inverse Gaussian OOD distributions". How do the authors make sure the real distribution fit in the designed piecewise linear function.
> >
> > [1] Towards optimal feature-shaping methods for out-of-distribution detection. ICLR 2024.
> >
> > ----------------------
> >
> > Update 04 Apr,
> >
> > Thank the authors for the detailed response! Most of my concerns have been addressed.

---

> > > ### Author Response · Authors · 2025-04-02
> > >
> > > *Thank you! Most of my concerns are solved, and thus, I raised my score to 3*
> > >
> > > **Response 1:** Thanks.
> > >
> > > *But I'm still confused with the …."*
> > >
> > > **Response 2:** Apologies – after reading the sentence in L344-345, we think that the way it was written may have led to confusion: it should be “The above distributions *result in optimal shaping functions* that all approximately fit …”  Thank you for pointing this out, we will change as follows:
> > > “The above distributions result in optimal shaping functions that all approximately fit in a piecewise linear function family as shown in Figure 4, so in the experimental section, we explore this family as shaping functions.”
> > >
> > > *In my understanding, the authors conduct experiments on  ...*
> > >
> > > **Response 3:** We would like to clarify that our method is not approximating OOD probability distributions using piecewise linear functions. Instead, our approach fits a very specific class of piecewise linear functions to the *feature shaping functions* that emerge from the optimization process (see Figure 2 that shows the shaping functions under the three OOD distribution assumptions).
> > >
> > >
> > > *First, a piecewise linear function can fit any functions, as long as there are enough number of “pieces”. [1] already uses a 100-piece function ... What is the advantage … ?*
> > >
> > > **Response 4:**
> > >
> > > Note we are not suggesting that simply using a piecewise linear function as a shaping function is new or in itself is our contribution.  The primary contributions of our work are the development of a theory for constructing OOD features, showing how the theory with specific distribution assumptions can explain SOA methods, and an example use of the theory by suggesting a specific family of shaping functions that can lead to better performance.
> > >
> > > There is similarity of our work with respect to the last contribution to [1] (and also other works e.g., ReAct, VRA) in the use of piecewise linear functions for shaping functions.  Note that our contribution is not to simply propose the use of piecewise linear functions – as that is too general a class, and as the reviewer rightly says, can approximate any function.  So just proposing the use of piecewise linear functions with many degrees of freedom (e.g., 100) would not in itself lead to any useful algorithm.
> > > The advantage of ours as compared to [1] is discussed in L81-108.  Our piecewise linear form is not the same as what [1] results in – they both come from entirely different loss functions:
> > > [1] optimizes over shaping functions:
> > >
> > > $\min_{ \{ f \text{  such that  } |f(z)/z|<K \} }  -\| \text{ mean of f given ID } – \text{ mean of f given OOD} \|$,
> > >
> > > To simplify the optimization from an infinite dimensional function optimization to a finite dimensional one, the problem is discretized so that theta(z) = f(z)/z is restricted to piecewise linear functions.
> > >
> > > Our loss function is
> > >
> > > $\min_{ p(\tilde z|z) } -KL(p(\tilde z|z)) + IB(p(\tilde z|z))$
> > >
> > > where in a simplified case we assume $p(\tilde{z}|z) \sim N(\mu(z),\sigma(z))$ where $\mu(z)$ is the mean of the shaping function.  We perform the infinite dimensional optimization over *general* functions $\mu(z), \sigma(z)$.  *Note that we are not restricting our shaping functions to be piecewise linear in our optimization,* while [1] does – so the class of shaping functions we optimize over is actually more general than the 100 piece (or any other number of pieces) functions in [1] that are optimized over.  Considering Gaussian, Laplace, IG OOD distributions, we show over all hyper-parameters (alpha, beta, distribution parameters) that the optimized function $\mu(z)$ can be expressed approximately by a very specific piecewise linear function family shown in Fig 4 (not *any* piecewise function).
> > >
> > > The advantages of our approach over [1] are two-fold.  First, in terms of theory, our approach explains the implicit assumptions in SOA methods.  Second, experimentally, our piecewise family out-performs [1].
> > >
> > >
> > > *Second, more importantly, OOD distributions are typically unknown in practice, ….*
> > >
> > > **Response 5:** We want to clarify again – we are not fitting OOD distributions with piecewise functions, rather we are approximating our optimized feature shaping functions with a *specific* class of piecewise linear functions.
> > >
> > > We agree that real world distributions may not fit the three example distributions – see the discussion for Reviewer 4N3p Response 2.  We are not advocating these distributional assumptions and hence not seeking to fit *all* real distributions.  Rather, the purpose of our work is to explain SOA methods, which have shown similar traits to our feature shaping functions under these three OOD distributions.  So while our piecewise shaping functions may not approximate shaping functions resulting from *all* real-world distributions, it is more general than the implicit assumptions made in existing SOA methods, without making too general distributional assumptions, which may fall outside the class of real OOD distributions.

---

### Official Review · Reviewer_Ghj6 · 2025-03-14

**Overall Recommendation:** 3

**Summary:**

The paper introduces a variational information-theoretic framework for OOD detection. It models OOD features as random variables by optimizing a loss function that balances KL divergence for feature separability and Information Bottleneck (IB) regularization for compactness.

**Claims And Evidence:**

Yes.

**Essential References Not Discussed:**

Related work appears to be discussed.

**Experimental Designs Or Analyses:**

Yes.

**Methods And Evaluation Criteria:**

Yes.

**Other Comments Or Suggestions:**

minor typos:
Section 3.1: "we make some simplifications to gain insights to our theory and approach." → Should be: "we make some simplifications to gain insights into our theory and approach."?

**Other Strengths And Weaknesses:**

Strengths:
The method presented in the paper improves upon a certain method (e.g. ReAct) by providing a theoretical framework rather than relying on heuristic rules like clipping activations. Unlike ReAct, which applies a fixed threshold to activations, the proposed approach learns an optimal OOD feature distribution using variational optimization. This allows the model to adapt dynamically to different OOD distributions, whereas ReAct may require manual tuning of the clipping threshold.

The paper’s method explains why different feature-shaping techniques (like ReAct, ASH, VRA, and FS-OPT) work under specific assumptions, offering a more general and theoretically grounded solution.

Empirically, it also achieves good results on benchmark datasets.

Weaknesses:

The method assumes some prior knowledge of the OOD distribution (e.g., Gaussian, Laplacian). How does it perform when the OOD distribution is completely unknown or highly complex?

The proposed optimization involves solving an infinite-dimensional problem using variational calculus. How does this impact computational efficiency compared to simpler heuristic methods like ReAct?

**Questions For Authors:**

Please see weaknesses part above.

**Relation To Broader Scientific Literature:**

This work builds upon and extends previous research by enhancing OOD detection, a crucial aspect of ensuring the reliability of deep learning models in real-world applications.

**Theoretical Claims:**

I have browsed the proof but have not examined it line by line.

---

> ### Author Rebuttal · Authors · 2025-03-31
>
> *Question 1: The method assumes some prior knowledge of the OOD distribution (e.g., Gaussian, Laplacian). How does it perform when the OOD distribution is completely unknown or highly complex?*
>
> **Response 1:**  Our experiments show the performance of our piecewise linear family that encompasses Gaussian, Laplacian, Inverse Gaussian perform well across datasets, even when the OOD distributions do not closely fit some of these datasets (see Reviewer 4N3p Response 2).  Our piecewise linear form makes more general distribution assumptions than SOA techniques as several SOA methods are related to our framework under one of the three distributions above, which is the reason for its out-performance of existing SOA.
>
> *Question 2: The proposed optimization involves solving an infinite-dimensional problem using variational calculus. How does this impact computational efficiency compared to simpler heuristic methods like ReAct?*
>
> **Response 2:** The inference cost is about the same as existing methods like ReAct. Since this optimization is done offline in training, the cost is not important.
>
> The complexity of optimization for 1D is $\mathcal O (NMK)$ where N is the samples of $p(z)$, M is the samples of $p(\tilde z|z)$ and $K$ is the number of gradient descent iterations.  In our experiments with $N=M=200$, the algorithm took between 5-10mins on a standard desktop.
>
> *Question 3: minor typos: Section 3.1: "we make some simplifications to gain insights to our theory and approach." → Should be: "we make some simplifications to gain insights into our theory and approach."?*
>
> **Response 3:** Thanks; will do.

---

### Official Review · Reviewer_4N3p · 2025-03-17

**Overall Recommendation:** 3

**Summary:**

The paper presents a novel theoretical framework for constructing out-of-distribution (OOD) detection features in neural networks using a variational information-theoretic approach. The key contribution is a novel loss functional that consists of a KL divergence term that maximizes the separation between in-distribution (ID) and OOD feature distributions, and an information bottleneck (IB) term that favors compressed features that retain relevant OOD information while discarding unnecessary details.

The loss functional is optimized using variational methods, leading to the derivation of OOD features as random variables. The mean of these random features corresponds to deterministic shaping functions used in existing OOD detection methods. By carrying large experiments, the authors show that their proposed piece-wise linear shaping function outperforms existing methods on standard OOD benchmarks, including ImageNet-1k and CIFAR datasets. The authors show that their method generalizes well across different architectures (e.g., ResNet, MobileNet, Vision Transformers) and datasets.

---
### Update after rebuttal

I thank the authors for their responses. I will maintain my initial score.

---

**Claims And Evidence:**

While the paper presents a strong theoretical framework with insightful connections to existing methods, certain claims, however, lack empirical validation, particularly those related to:

1. OOD distribution assumptions: The proposed theoretical framework assumes specific statistical distributions for in-distribution (ID) and OOD data (Gaussian, Laplacian, and Inverse Gaussian). Real-world OOD data may not necessarily follow these distributions, and no empirical evidence is provided to justify these assumptions. The method's generalization to arbitrary OOD distributions is unclear.

2. Superiority of random features over deterministic ones: The paper claims that random features (rather than deterministic shaping functions) are more optimal for OOD detection. However, this is only based on the assumed specific statistical distributions where the choice of the parameters seems handpicked and the dimensionality of the probability space reduced to 1D.

3. Necessity of the Information Bottleneck term: The IB term is used for regularization, ensuring that features retain only the necessary OOD information while being compressed. However, the necessity of the IB term is not experimentally tested. Put differently, we don't know how good the loss functional would be without the IB term. An ablation study could help determine the effectiveness of the loss.

4. Scalability to high-dimensional data: The optimization problem is inherently infinite-dimensional, requiring numerical approximation techniques. While they propose a computationally feasible method in 1D, the complexity of extending it to high-dimensional feature spaces is not addressed. Unfortunately, this is discussed extensively in the paper, even the complexity of the approach in a 1D setting is not discussed.

**Essential References Not Discussed:**

The paper introduces a variational formulation for OOD feature extraction but fails to cite prior works that have applied variational or information-theoretic techniques to OOD detection. There are existing variational formulations for OOD detection that should be referenced for proper context.

[1] proposes an information-theoretic approach for OOD detection by using likelihood ratios instead of raw likelihoods. [1] is similar to this paper in the sense that it formulates OOD detection from an information-theoretic perspective. The likelihood ratio idea relates to KL divergence-based feature separation in the current work. It would be useful to know how the KL divergence optimization proposed in this paper compares to likelihood-ratio-based OOD detection.

[2] Examines the limitations of information-theoretic methods like normalizing flows for OOD detection. Since the proposed method is information-theoretic, the authors should discuss known pitfalls from this prior work. For instance, does the variational approach in this paper avoid the same failure modes?

[3] Investigates how pre-trained models learn OOD features, showing that deeper layers contain more useful information. This work suggests that OOD detection improves when using deep feature representations. The current paper should cite this when justifying why it focuses on feature-based OOD detection.

References:
[1] Ren et al. (2019) – "Likelihood Ratios for Out-of-Distribution Detection" (NeurIPS 2019)
[2] Kirichenko et al. (2020) – "Why Normalizing Flows Fail for Out-of-Distribution Detection" (NeurIPS 2020)
[3] Fort et al. (2021) – "Exploring the Limits of Out-of-Distribution Detection" (NeurIPS 2021)

**Experimental Designs Or Analyses:**

Yes, I examined the experimental design and analyses focusing on methodology soundness, validity of comparisons, and potential issues. Below is a structured review.

Strengths: The authors compare their method to several state-of-the-art (SoA) OOD detection approaches, including:
feature shaping methods (e.g., ReAct, FS-OPT, VRA-P, ASH), Softmax-based methods (e.g., MSP, ODIN), and energy-based methods (e.g., Energy, DICE). This makes their results contextualized within the broader OOD detection field.

Their experiments were conducted on widely accepted datasets. They considered ImageNet-1k as the in-distribution (ID) dataset, and
Species, iNaturalist, SUN, Places, etc. as the OOD datasets. They considered CIFAR-10/100 as ID datasets with common OOD datasets such as TinyImageNet, SVHN, Texture, Places365, LSUN-Cropped, LSUN-Resized, iSUN, CIFAR100/10.  All these datasets seem appropriate for natural image distribution shifts, making them relevant for evaluating OOD detection.

The paper evaluates many widely accepted neural networks models using metrics such False Positive Rate at 95% True Positive (FPR95) for OOD detection, and AUROC to measure overall separability between ID and OOD samples. These are standard and valid metrics for OOD detection performance evaluation.

**Methods And Evaluation Criteria:**

Yes, the proposed methods and evaluation criteria mostly make sense for the problem of out-of-distribution (OOD) detection. The models and datasets they considered are widely used in the OOD literature as benchmarks.

**Other Comments Or Suggestions:**

None.

**Other Strengths And Weaknesses:**

Overall, the paper is well-written and insightful. Providing answers to the questions I asked above would bringing more clarity about the key contributions of the paper.

**Questions For Authors:**

Key questions:

1. How does the variational optimization framework compare to likelihood ratio-based OOD detection (e.g., Ren et al., NeurIPS 2019)? This method maximizes KL divergence to separate ID and OOD feature distributions. Likelihood ratio-based OOD detection also exploits divergence between ID and OOD distributions but does so directly in a Bayesian framework. Understanding the difference in theoretical properties would clarify whether this approach is more generalizable or computationally efficient than likelihood-based OOD methods.

2. What specific assumptions about OOD distributions are necessary for the proposed feature shaping functions to be optimal? This method assumes Gaussian, Laplacian, or Inverse Gaussian distributions for OOD features. However, real-world OOD data may not follow these assumptions. It is unclear how the method would generalize to cases where OOD features do not fit these distributions.

3. Why didn't the authors include an ablation study comparing performance with and without the Information Bottleneck (IB) term? The IB term is a key part of the proposed framework, but there is no direct evidence that it improves results. Without an ablation study, it remains unclear whether IB contributes meaningfully or if similar results could be achieved without it.

4. How were the hyper-parameters (e.g., IB weight $\alpha$) selected, and were they tuned consistently across all datasets?

5. Since the use case that was presented in the paper focuses on 1D data, I am curious to know how well this method scales to high-dimensional data (e.g., Vision Transformers, large-scale multi-modal datasets)?

**Relation To Broader Scientific Literature:**

OOD detection is a widely studied problem in machine learning, and various techniques have been proposed, including confidence-based, energy-based, distance-based, and feature-shaping methods. The paper's contributions build upon these methods while introducing a variational formulation using information theory.

With respect to Confidence-based OOD Detection methods, the paper moves away from softmax-based methods, which rely heavily on the output space rather than feature space. The authors argue that feature-based approaches offer better generalization than confidence-based methods.

For Energy-based OOD Detection methods, the variational loss function in this paper can be seen as an extension of energy-based approaches, since KL divergence maximization between ID and OOD features implicitly encourages separation. However, the proposed method focuses on shaping feature distributions rather than directly modifying the energy function.

With respect to Distance-based OOD Detection methods, the proposed method does not rely on distance metrics, but instead optimizes a variational loss functional to encourage separation between ID and OOD distributions. However, the KL divergence term serves a similar function to distance-based methods by ensuring OOD features are statistically different from ID features.

For Information-Theoretic OOD Detection methods, the proposed loss explicitly incorporates IB regularization into the OOD feature optimization process. The authors justify several existing OOD shaping methods (e.g., ReAct, FS-OPT) as special cases under their framework.

**Theoretical Claims:**

I didn't check in depth the proofs of the theoretical claims. In fact, I mostly focused on understanding the proposed loss, the variational framework, and its effectiveness in detecting OOD samples.

---

> ### Author Rebuttal · Authors · 2025-03-31
>
> Key Question 1
>
> **Response 1**: Thanks for the ref, we’ll cite.  We agree with the use of likelihood ratio (LR) as a score over the likelihood.  However, our work focuses on feature shaping not scoring (L60-61, 2nd column).  Our work separates the distributions of the resulting OOD feature shape distributions under ID/OOD through the KL divergence (& IB) to derive optimal shaping features, while [1] uses the LR of the original neural net feature as a score.  So while the LR relates to the KL div. (as the expectation of the LR), [1] defines a score, whereas our approach determines feature shape.
>
> Key Question 2
>
> **Response 2:** Our theoretical framework does not assume specific distributions: it is applicable to any ID/OOD distribution - no assumptions are needed for optimality.
>
> To concretely illustrate the theory, example distributions are explored.  Section 4 explains our choice of distributions: Gaussian is common in probabilistic analysis; Laplace models heavy-tailed outliers typical of OOD; and Inverse Gaussian serves as a reasonable assumption when no prior knowledge is known - large when ID distribution is low and vice versa. These examples reveal that current SOA heuristics may implicitly assume properties of these distributions.  Having said that, our examples have similarities to distributions consistent with OOD data in the real-world.  VRA (Xu et al.) plots distributions of various OOD data in the ImageNet benchmark – OOD qualitatively have heavier tails and sharper peaks than ID data.  Gaussian ID / Laplace OOD choice match these properties.  In e.g. VIT-B-16, the distributions ID/OOD appear Gaussian – see https://drive.google.com/file/d/1XANDdWfXdy4MONY0MnQfaVOBoVNy7r0a/view?usp=sharing
> In practice, OOD data will change based on application so it isn’t desirable to match too closely to any real-world dataset otherwise it would not generalize.
>
> Key Question 3
>
> **Response 3:** Ablations are not shown as, without IB, the loss is ill-posed (L114-116, 2nd column) - the optimization does not converge since distributions can be arbitrarily separated, increasing the KL term indefinitely. Appendix A explains this (L153-159, 2nd column) with a case that can be analyzed analytically - linear mean feature with Gaussian ID/OOD distributions. Without IB, the slope becomes unbounded. In the more general non-linear case, which cannot be solved analytically, the optimization also diverges without IB. This is consistent with literature – FS-Opt and VRA that separate distributions (though through mean separation) impose constraints by effectively restricting the slope of the feature shaping function. While restricting the slope does produce a well-posed problem, it may not be addressing the underlying cause of the ill-posed-ness. IB is more naturally suited to the OOD problem – retaining OOD information while compressing the feature.
>
> Key Question 4
>
> **Response 4:** See Section 5 Lines 342-350 (2nd column). Yes, consistently chosen over all datasets.
>
> Key Question 5
>
> **Response 5:** Our method has been demonstrated on high-dim data - the shaping functions are applied on each component of the penultimate layer feature (Sec. 5), as SOA does.  We experimented with two vision transformers (Table 1: ViT-B-16 and ViT-L-16).
>
> *Superiority of random features over deterministic …*
>
> **Response 6:** We formulated general random features; a question is whether they result in a better loss.  The answer is yes - maybe not for all cases, but all cases that we examined (all the distributions and hyperparameters in all the plots shown have non-zero standard deviation, we didn’t show them because of space and didn’t find it insightful).  We will clarify the statement in the paper – that our intent is not to claim random features are always more optimal, just that there is reason to consider them.
>
> *Scalability to high-dimensional data …*
>
> **Response 7:**  Our current methodology applies to high-dim data with independence assumptions (L185-190) as SOA.  Generalization of the independence assumptions is challenging as numerical methods presented won't scale - grid representations of multi-d probabilities are infeasible. While we have preliminary ideas to address this, it is future work. The current paper's goal is to establish foundational theory and demonstrate its utility. Despite focusing on 1D shaping functions, consistent with SOA, our theory successfully explains these methods and predicts new shaping functions. This underscores the promise of our approach and lays groundwork for future exploration.
>
> See Reviewer Ghj6 Response 2 for complexity.
>
> *[2] limitations of information-theoretic methods like normalizing flows*
>
> **Response 8:**[2] shows why normalizing flows that produce exact likelihoods do not work well as a score for OOD detection. [2] relates to the scoring function, not feature shaping – which is the focus of our work. We’ll cite [2] in our discussion on scores.
>
> *[3]…should cite*
>
> **Response 9:** Thanks; will do.

---

> > ### Comment · Reviewer_4N3p · 2025-04-06
> >
> > I thank the authors for their efforts. Their responses seem convincing enough. I will maintain my score.

---

### Decision · Program_Chairs · 2025-05-01

**Decision:**

Accept (poster)

**Comment:**

This paper proposes a novel theoretical framework for out-of-distribution (OOD) detection in neural networks. Reviewers generally acknowledge the novelty of the proposed framework and commend several positive aspects, particularly its innovative theoretical perspective and demonstrated experimental efficacy. However, the reviewers also raise several critical concerns that warrant attention: (1) insufficient empirical validation of the key theoretical assumptions underlying the framework; (2) insufficiently clarified connections between the proposed methodological implementations and their corresponding theoretical analyses. Notwithstanding these limitations, the work constitutes a substantive contribution to the International Conference on Machine Learning (ICML) research community. The authors demonstrate strong potential to address the identified issues through systematic revisions. Therefore, a moderate-priority acceptance is recommended, provided that the authors adequately address the aforementioned concerns in their revision.